# N-Acetyltransferase 10 represses Uqcr11 and Uqcrb independently of ac4C modification to promote heart regeneration

Wenya Ma[1,2,3,8], Yanan Tian[1,2,8], Leping Shi[1,2,8], Jing Liang [4], Qimeng Ouyang[1,2], Jianglong Li[1,2], Hongyang Chen[1,2], Hongyue Sun[1,2], Haoyu Ji[1,2], Xu Liu[1,5], Wei Huang[1,2], Xinlu Gao[1,2], Xiaoyan Jin[1,2], Xiuxiu Wang[1], Yining Liu[1], Yang Yu[1], Xiaofei Guo[1], Ye Tian[6], Fan Yang[2], Faqian Li[7], Ning Wang[2] & Benzhi Cai [1,2,3] ✉

Translational control is crucial for protein production in various biological contexts. Here, we use Ribo-seq and RNA-seq to show that genes related to oxidative phosphorylation are translationally downregulated during heart regeneration. We find that Nat10 regulates the expression of Uqcr11 and Uqcrb mRNAs in mouse and human cardiomyocytes. In mice, overexpression of Nat10 in cardiomyocytes promotes cardiac regeneration and improves cardiac function after injury. Conversely, treating neonatal mice with Remodelin−a Nat10 pharmacological inhibitor−or genetically removing Nat10 from their cardiomyocytes both inhibit heart regeneration. Mechanistically, Nat10 suppresses the expression of Uqcr11 and Uqcrb independently of its ac4C enzyme activity. This suppression weakens mitochondrial respiration and enhances the glycolytic capacity of the cardiomyocytes, leading to metabolic reprogramming. We also observe that the expression of Nat10 is downregulated in the cardiomyocytes of P7 male pig hearts compared to P1 controls. The levels of Nat10 are also lower in female human failing hearts than non-failing hearts. We further identify the specific binding regions of Nat10, and validate the pro-proliferative effects of Nat10 in cardiomyocytes derived from human embryonic stem cells. Our findings indicate that Nat10 is an epigenetic regulator during heart regeneration and could potentially become a clinical target.

Ischemic heart disease (IHD) remains one of the leading causes of death worldwide, with the loss of cardiomyocytes being the primary contributor to cardiac remodeling and heart failure. Therefore, replenishing the loss of cardiomyocytes is crucial for the prevention and treatment of IHD. Since researchers have discovered that the cardiomyocytes of newborn hearts, as opposed to those of adult mammals, possess the potential to regenerate, stimulating endogenous cardiomyocyte proliferation has emerged as an appealing strategy

[1]Department of Pharmacy at the Second Affiliated Hospital, Harbin Medical University, Harbin, China. [2]Department of Pharmacology at College of Pharmacy (National Key Laboratory of Frigid Zone Cardiovascular Diseases, Key Laboratory of Cardiovascular Research, Ministry of Education), Harbin Medical University, Harbin, China. [3]Institute of Clinical Pharmacy, NHC Key Laboratory of Cell Transplantation, the Heilongjiang Key Laboratory of Drug Research, Harbin Medical University, Harbin, China. [4]Institute of Hematology & Blood Diseases Hospital, Chinese Academy of Medical Sciences & Peking Union Medical College, Tianjin, China. [5]Department of Laboratory Medicine at The Fourth Affiliated Hospital, Harbin Medical University, Harbin, China. [6]Department of Pathophysiology and the Key Laboratory of Cardiovascular Pathophysiology, Harbin Medical University, Harbin, China. [7]Department of Laboratory Medicine and Pathology, University of Minnesota, Minneapolis, MN, USA. [8]These authors contributed equally: Wenya Ma, Yanan Tian, Leping Shi. ✉e-mail: caibz@ems.hrbmu.edu.cn

for promoting cardiac regeneration and repair. Some genes and non-coding RNAs have been identified as the regulators of heart repair and regeneration[1–6], which helps us gain a much deeper insight into this process. The transition of oxygen-rich environment and the metabolism from glycolysis to mitochondrial oxidative phosphorylation (OXPHOS) after birth are responsible for the cell-cycle arrest of cardiomyocytes[7,8]. Despite this understanding, the regulatory genes and precise mechanisms that govern this metabolic reprogramming process have yet to be fully elucidated. Mitochondrial OXPHOS relates to electron transfer chains containing mitochondrial respiratory chain complexes (MRCC) I-V. Among them, MRCC III is a necessary protein for mitochondrial OXPHOS, and MRCC V is closely related to ATP synthesis. However, the involvement of mitochondrial OXPHOS in regulating cardiomyocyte proliferation is still unclear.

Various genes regulating cardiomyocyte renewal and heart regeneration have been identified through the analysis of "omics" - methods based on a variety of technologies[9–12]. Despite the prevalence of gene expression profile analyses in contemporary research, the emphasis is primarily on transcriptional levels, with limited attention given to translational profiling during heart regeneration. Ribo-seq reveals translational regulation by profiling the number and position of ribosomes on mRNA through recognizing ribosome protective fragments (RPFs)[13,14]. It has been applied to explore translational regulation in various biological processes[15–20]. However, translational regulation of cardiac regeneration has not yet been documented. Given that translational regulation is the primary determinant of protein production, a critical aspect of gene expression, it is imperative to investigate translatomic changes during cardiac regeneration to unravel the key regulators. N-Acetyltransferase 10 (Nat10) is a nuclear protein with acetyltransferase activity which is rich in lysine and RNA cytidine. It has been reported that Nat10 regulates the RNA translation and the behaviors of various cancer cells by enriching the ac4C modification of mRNAs[21–25]. Besides, Nat10 is involved in piRNA-HAAPIR regulation of cardiomyocyte apoptosis by enhancing the ac4C modification of Tfec mRNA[26]. However, the role and underlying mechanism of Nat10 in cardiomyocyte proliferation and heart regeneration remains unclear.

Here, we conduct translatomics research on regenerative heart tissue to profile the translational landscape and explore the key regulators involved in heart regeneration. Our findings indicate that the translation of genes related to OXPHOS is suppressed during cardiac regeneration. The OXPHOS-related genes Uqcr11 and Uqcrb have the function of regulating cardiomyocyte cycling. We also identify Nat10 as a key regulator that non-classically represses the translation of Uqcr11 and Uqcrb independently of ac4C modification, which induces mitochondrial metabolic reprogramming and promotes cardiomyocyte proliferation. We further demonstrate that overexpression of Nat10 in cardiomyocytes enhances cardiomyocyte proliferation and improves heart function after myocardial infarction (MI). Conversely, down-regulation of Nat10 through genetic or inhibitory approaches inhibits cardiomyocyte cycling and heart regeneration after injury in neonatal mice. We further identify the specific binding regions of Nat10 to Uqcr11 and Uqcrb mRNAs and reveal transcriptional factor Hes1 as the upstream regulator of Nat10. Collectively, our findings suggest Nat10 as a key epigenetic regulator mediating heart regeneration and a promising therapeutic target for enhancing cardiomyocyte cycling and improving heart function after MI.

## Results

### Ribosome profiling of regenerative heart tissues

To gain insights into the translational landscape involved in cardiac regeneration, we performed apical resection (AR) on newborn mice to induce cardiac regeneration. The heart tissues in the damaged marginal area were collected for ribosome profiling (Ribo-seq) on the 6th day after AR, and transcriptomic changes in the regenerated hearts were revealed by RNA-seq (Fig. 1a). The analysis of Ribo-seq data

showed that the majority of ribosome-protected fragments (RPFs) mapped to the coding sequence (CDS) regions of genes, with the fewer fragments found in the 5' and 3' untranslated regions (Supplementary Fig. 1a). The size distribution of the RPFs was consistent with the expected pattern, with a peak at 28 nucleotides (nt) (Supplementary Fig. 1b). Furthermore, the ribosome footprint positions analysis showed that RPFs were predominantly located in the first reading code frame and exhibited a 3-nt periodicity along the transcript, and ribosomes were evenly distributed throughout the CDS region with good quality (Supplementary Fig. 1c).

Upon analyzing the Ribo-seq data, it was found that the RPF counts of 486 genes manifested alterations. Specifically, 258 of these genes underwent up-regulation while 228 were down-regulated (Fold Changes>1.5, $P < 0.05$) (Fig. 1b, c). KEGG analysis of the altered RPFs revealed that extracellular matrix (ECM) receptor interaction pathway was the most significantly enriched, suggesting that the number of ribosomes translating mRNA of ECM-related genes was primarily affected (Fig. 1d). This is consistent with previous studies that uncovered the role of ECM in controlling cardiac repair and regeneration[9,27,28]. Furthermore, RNA-seq analysis showed that 567 transcripts increased while 327 transcripts decreased (Fig. 1e, f). Among these, the up-regulation of Zeb2, known to promote angiogenesis after heart injury[29], and the down-regulation of Col1, a marker of cardiac fibrosis[30], were observed. KEGG analysis of the transcriptomics revealed the enrichment of MAPK signaling pathway and Rap1 signaling pathway, both associated with cardiac repair (Fig. 1g)[31–33].

To better know the correlation between RPFs and RNA, we analyzed the data from Ribo-seq and RNA-seq analyses and identified 8383 genes common to both datasets (Fig. 1h). Among these genes, approximately 93% exhibited no or minimal changes in both transcription and RPF levels (Fig. 1i, j). By contrast, 57 genes showed significant changes both transcriptionally and translationally. Of these, 40 genes changed consistently in both RNA and RPF levels, of which 30 were up-regulated and 10 were down-regulated, while 17 genes changed contradictorily. Besides, 548 genes merely changed in RNA level, while 437 genes changed exclusively in RPF level (Fig. 1i, j). Altogether, it indicates that the majority of cardiac genes exhibit unaltered RNA and RPF levels during heart regeneration, with only some genes being regulated at the either transcriptional or translational level. Furthermore, the changes at the two levels appears to be relatively independent.

### Dynamic translational regulation reveals oxidative phosphorylation-related genes are down-regulated during heart regeneration

We further studied the translation efficiency (TE) of genes to depict the dynamics of translational regulation during cardiac regeneration. TE is important to reflect the changes in the translatomics landscape, which is associated with RNA quantity and RPF density. Therefore, the data from Ribo-seq and RNA-seq were analyzed to evaluate TE. The results showed that the TE of 328 genes increased, while the TE of 586 genes decreased (Fig. 2a, b). The pathways enriched in RNA and RPF data were also observed in TE analysis (Supplementary Fig. 2a). In addition, ErbB signaling pathway, mTOR signaling pathway, and VEGF signaling pathway were enriched, which have been previously implicated in the regulation of cardiac regeneration[34–37].

Given the relationship between TE and the abundance of RNA and ribosomes on transcripts, TE fluctuations do not completely secure protein fluctuation. Therefore, to identify genes that undergo changes due to translational regulation during cardiac regeneration, we integrated and analyzed RNA, RPF, and TE data. It was revealed that 71 genes changed translationally but not transcriptionally (Fold changes >1.5, $P < 0.05$), including 25 up-regulated genes and 46 down-regulated genes (Fig. 2c). KEGG analysis showed that these genes were closely related to oxidative phosphorylation (OXPHOS) and protein digestion and absorption (Fig. 2d). Actually, OXPHOS-related genes were down-

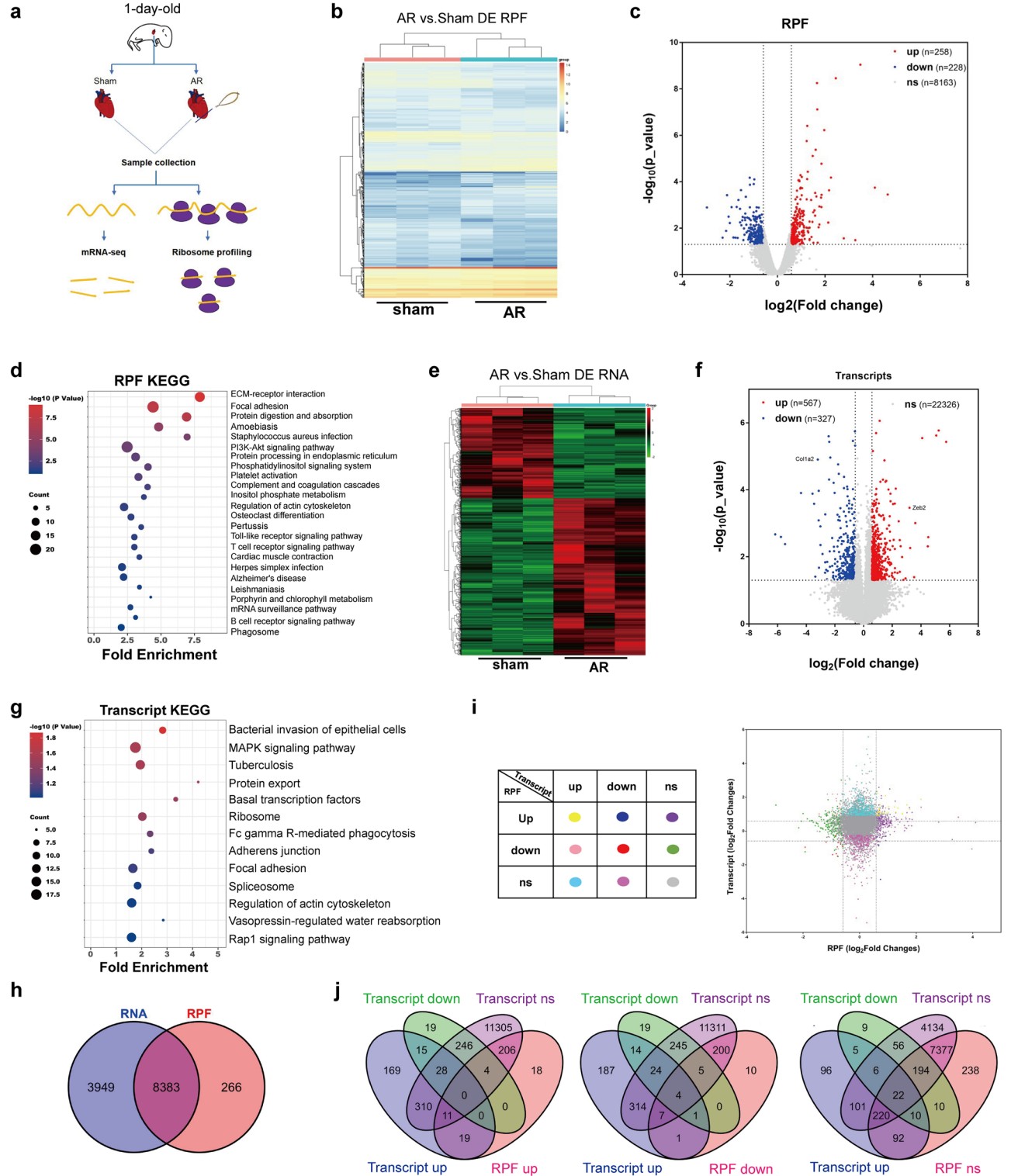

**Fig. 1 | Translatome and transcriptome profiling of regenerative hearts.**
**a** Schematics of the experimental approach generated using Adobe Illustrator and Microsoft PowerPoint. **b**, **c** Heatmap of differentially expressed RPFs (left) and Volcano plot of RPFs (right) in heart tissues. likelihood ratio test. **d** KEGG analysis of differentially expressed RPFs. Fisher exact test. **e**, **f** Heatmap of differentially expressed transcripts (left) and Volcano plot of transcripts (right) in heart tissues. F test. **g** KEGG analysis of differentially expressed transcripts. Fisher exact test. **h** The overlapping of RPFs and RNA. **i**, **j** The correlation analysis of RPF and RNA. Source data are provided as a Source Data file.

regulated and the protein digestion- and absorption-related genes were up-regulated translationally (Fig. 2e and Supplementary Fig. 2b). After birth, cells gradually rely on mitochondrial oxidative phosphorylation as their main metabolic approach, which is related to the loss of heart regeneration ability[7,8]. Thus, we focused on OXPHOS-related

genes Uqcr11, Uqcrb, and Atp5j2, which were regulated translationally (Fig. 2f).

In order to validate the sequencing data, the expression of Uqcr11, Uqcrb, and Atp5j2 was detected. We found no discernible variation in mRNA expression levels of these genes between the sham and AR

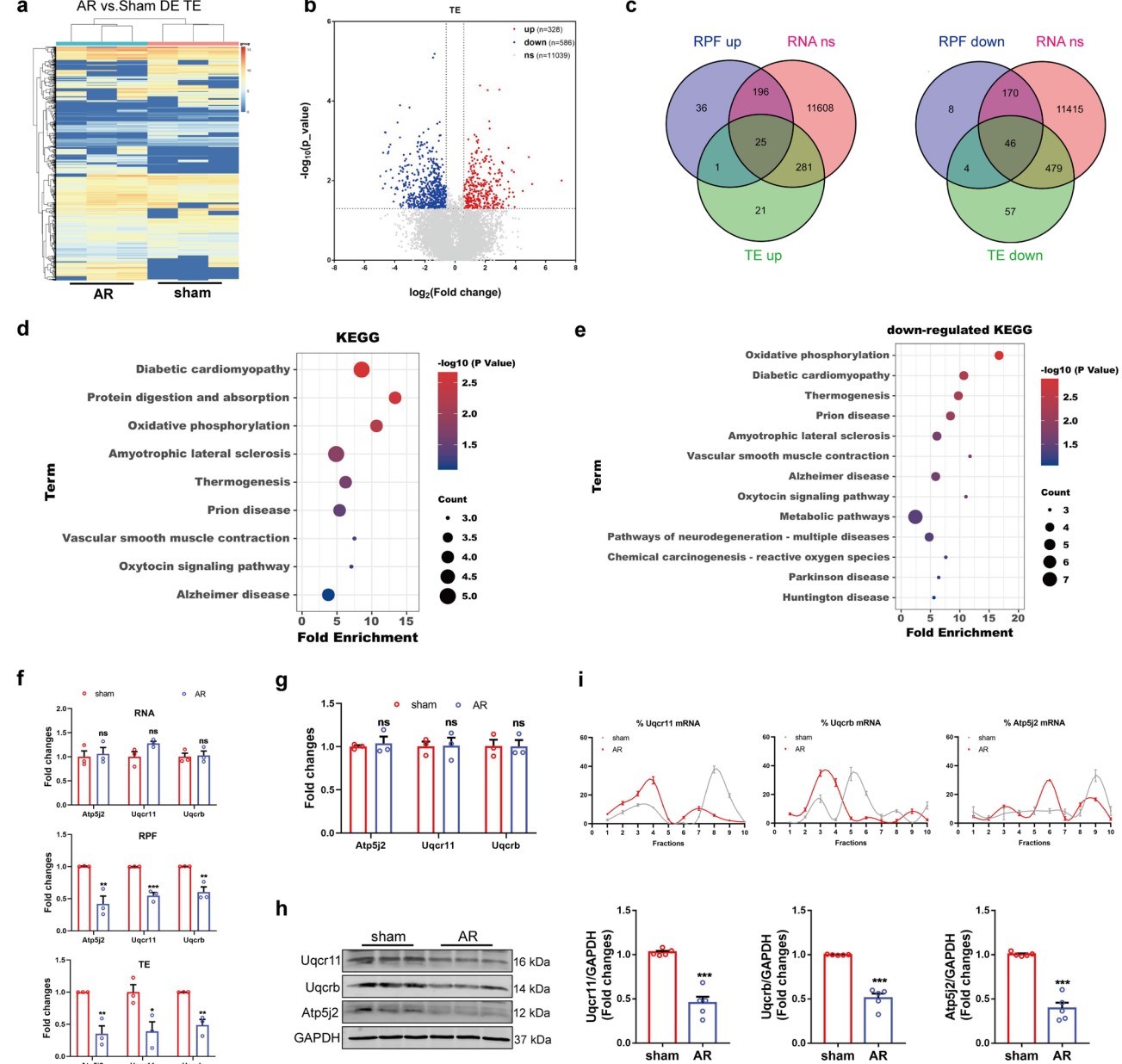

**Fig. 2 | The translational regulation in cardiac regeneration. a** Heatmap of differentially expressed TE. **b** Volcano plot of TE in heart tissues. A statistical test based on the H0 and H1 model fitting. **c** Venn diagram showing the overlap of genes of differentially expressed RPF and TE, and unchanged RNA. **d** KEGG analysis of genes with translational regulation by DAVID and HIPLOT. Fisher exact test. **e** The KEGG analysis of genes with down-regulated translation. Fisher exact test. **f** The fold changes of RNA (FPKM, F test), RPF (CPM, likelihood ratio test) and TE (FPKM, a statistical test based on the H0 and H1 model fitting) of Uqcr11, Uqcrb and Atp5j2 in

tissues (Fig. 2g). However, when we measured the protein expression of these genes in the heart tissues of both groups, we observed the down-regulation of Uqcr11, Uqcrb, and Atp5j2 in the AR group compared to the sham group (Fig. 2h). Subsequently, heart tissues of sham and AR mice were collected and subjected to polysome profiling analysis to investigate the translational regulation of these genes. Our findings revealed that more mRNA of Uqcr11, Uqcrb, and Atp5j2 were enriched in the light polysome fractions, while fewer mRNAs were enriched in the heavy polysome fractions in the AR mice compared to the sham mice (Fig. 2i and Supplementary Fig. 3a). These results show that translational regulation occurs during cardiac regeneration and

RNA-seq and Ribo-seq data ($n = 3$ independent mice). **g, h** The mRNA and protein expression level of Uqcrb, Uqcr11 and Atp5j2 in AR heart tissues (**g**, $n = 3$ independent mice; **h** $n = 5$ independent mice). Two-tailed Student's $t$ test with **g** or without Welch's correction **h**. **i** The translational regulation of Uqcrb, Uqcr11 and Atp5j2 in AR heart tissues analyzed by polysome profiling ($n = 3$ independent mice). *$p < 0.05$, **$p < 0.01$, ***$p < 0.001$ vs. sham group, ns, not significant. Data are presented as mean ± SEM. Source data are provided as a Source Data file. Exact $p$ values are provided in the Source Data file.

that the translation of OXPHOS-related genes Uqcr11, Uqcrb, and Atp5j2 is down-regulated, indicating that the energy source and metabolic mode may have undergone translational remodeling.

### Translation regulation of Uqcr11 and Uqcrb by Nat10

To explore the key epigenetic regulators of Uqcr11, Uqcrb, and Atp5j2, we analyzed the genes involved in translational regulation in the RNA-seq data (Fig. 3a). Subsequently, we analyzed the interaction between these genes and Uqcr11, Uqcrb, and Atp5j2. The top 10 genes were then selected and subjected to qRT-PCR verification (Fig. 3b, c). The results showed that the expression of Nat10, Alkbh5, Ythdf3, Mtor, and Gcn1

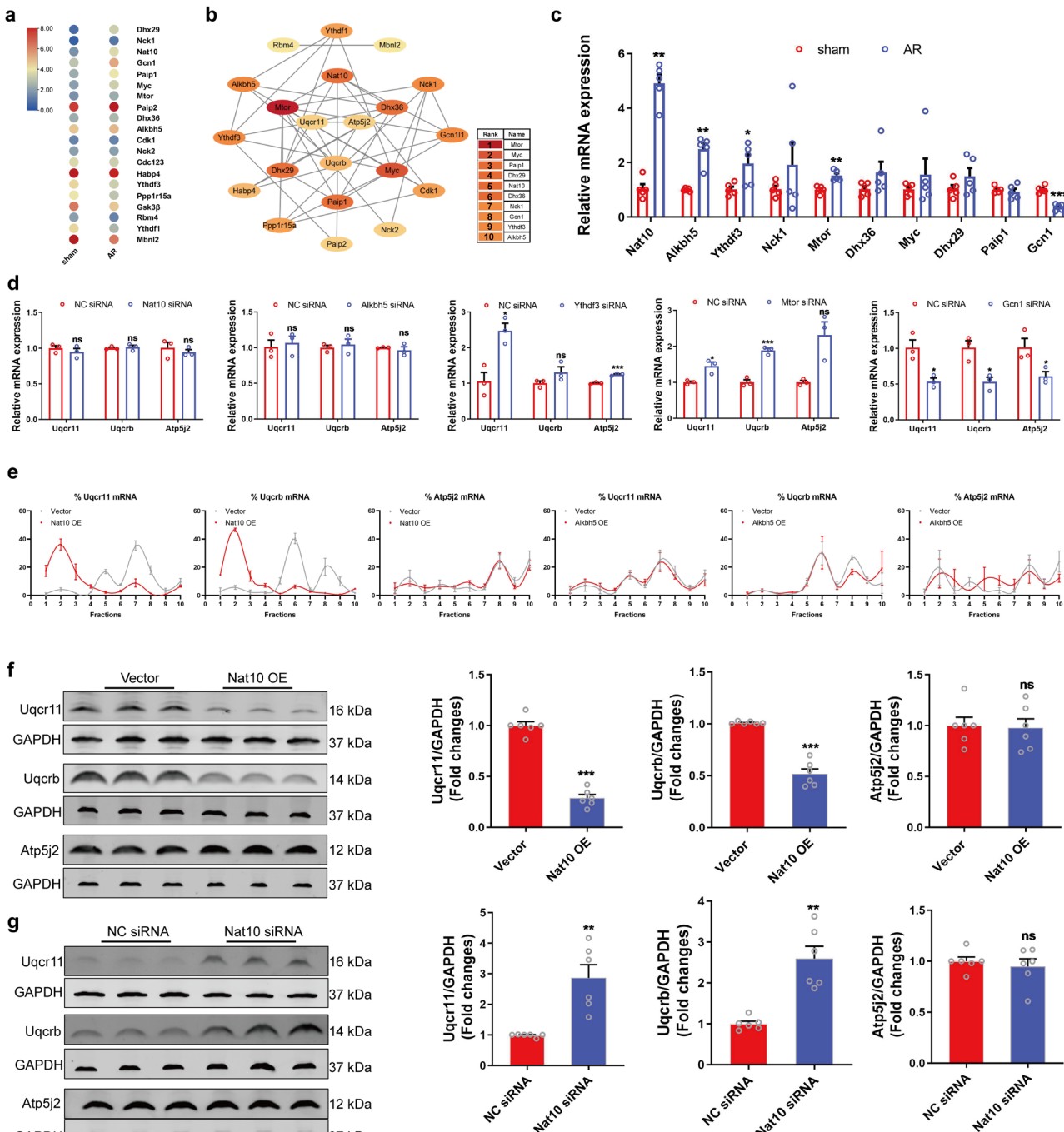

**Fig. 3 | The regulatory role of Nat10 in the translation of OXPHOS-related genes. a** Heatmap of genes involved in translational regulation in the RNA-seq data of AR mouse hearts. **b** A interaction network was constructed using STRING and visualized with Cytoscape (v. 3.7.1). **c** The expression of translation-related genes measured by qRT-PCR (n = 5 independent mice). *p < 0.05, **p < 0.01, ***p < 0.001 vs. sham group. **d** The effects of Nat10, Alkbh5, Paip2, Mtor, and Gcn1 on the mRNA expression of Uqcrb, Uqcr11 and Atp5j2 measured by qRT-PCR (n = 3 independent experiments). *p < 0.05, ***p < 0.001 vs. NC siRNA. **e** Polysome profiling analysis of Uqcrb, Uqcr11 and Atp5j2 in Nat10-overexpressed cardio-myocytes (n = 4 independent experiments). **f** and **g** Western blot analysis of Nat10, Uqcrb, Uqcr11 and Atp5j2 (n = 6 independent experiments). **p < 0.01, ***p < 0.001 vs. Vector or NC siRNA. Data were analyzed using Two-tailed Student's t test with or without Welch's correction for parameter analysis, Mann–Whitney two-tailed test for nonparametric analysis **c, d, f**, and **g**. Data are presented as mean ± SEM. Source data are provided as a Source Data file. Exact p values are provided in the Source Data file.

changed in the tissues of AR mice compared with those of sham mice, with Nat10 exhibiting the more significant up-regulation (Fig. 3c). We then detected the effects of Nat10, Alkbh5, Ythdf3, Mtor, and Gcn1 on the expression of Uqcr11, Uqcrb, and Atp5j2 mRNA and discovered that Ythdf3, Mtor, and Gcn1 affected the mRNA expression of Uqcr11, Uqcrb, and Atp5j2, while Nat10 and Alkbh5 did not change their expression (Fig. 3d). This indicates that Ythdf3, Mtor, and Gcn1

regulate the transcription of these genes, while Nat10 and Alkbh5 have no effect. To explore translational regulators of OXPHOS-related genes during cardiac regeneration, we applied polysome profiling analysis and found that Nat10 decreased Uqcr11 and Uqcrb mRNA enrichment in the heavier polysome fractions, shifting their distribution from the heavier to the lighter polysome fractions whereas showed no effect on the distribution of Atp5j2 mRNA (Fig. 3e and Supplementary Fig. 3b).

Meanwhile, Alkbh5 had no effect on their translation (Fig. 3e). Western blot results showed that overexpression of Nat10 reduced the protein expression levels of Uqcr11 and Uqcrb, while knockdown of Nat10 enhanced their protein expression levels (Fig. 3f, g and Supplementary Fig. 4). When the protein degradation was inhibited by MG132, a proteasome inhibitor, Nat10 overexpression still could inhibit the Uqcr11 and Uqcrb protein levels (Supplementary Fig. 5). The results described above indicate that Nat10 is the key regulator of the translation of Uqcr11 and Uqcrb. Overexpression of Nat10 inhibits the translation of Uqcr11 and Uqcrb, decreasing their expression; conversely, knockdown of Nat10 increases their expression.

### The translation of Uqcr11 and Uqcrb mediating cardiomyocyte proliferation is regulated by Nat10 in an ac4C-independent manner

Nat10, the only known RNA ac4C modifying enzyme, is pivotal in various processes. To unveil the mechanism of Nat10 regulating gene translation, we applied a predictor[38] of ac4C sites and conducted ac4C RIP to identify the ac4C modification on the mRNA of Uqcr11 and Uqcrb. Unexpected, the predicted and ac4C RIP results showed that there was no ac4C modification on the mRNA of Uqcr11 and Uqcrb (Fig. 4a and Supplementary Fig. 6). It suggests that Nat10 probably regulates the expression of Uqcrb and Uqcr11 not through a classic approach of regulating their ac4C modification and translation in ribosomes. Interestingly, Nat10 RIP results revealed that Nat10 could bind to the mRNA of Uqcr11 and Uqcrb directly (Fig. 4b). The mRNA traffic and localization are associated with translational expression of genes. Our investigation further uncovered that Nat10 affected the RNA localization of Uqcr11 and Uqcrb. Overexpression of Nat10 increased the nuclear distribution of Uqcr11 and Uqcrb mRNAs, and Nat10 knowdown promoted their cytoplasmic distribution (Fig. 4c and Supplementary Fig. 7). Moreover, RIP experiment showed that overexpression of Nat10 decreased the abundance of Uqcr11 and Uqcrb mRNAs pulled down by Nxf1 which is the main 'transport vehicle' for mRNAs (Supplementary Fig. 8). This indicates that Nat10 binds to Uqcr11 and Uqcrb respective transcripts, inhibits their mRNA nuclear export, and reduces the number of their mRNA translated in ribosomes, which exerts its inhibitory effect on Uqcr11 and Uqcrb mRNA translation in a non-classical manner.

Additionally, to further identify the specific binding of Nat10 to Uqcr11 and Uqcrb mRNAs, we predicted the binding sites between Nat10 and Uqcr11 or Uqcrb using the catRAPID algorithm (Supplementary Fig. 9A and B). Then, we constructed the Nat10 plasmids with binding site mutations, as well as Uqcr11 and Uqcrb mRNA mutations with biotin labeled. The RIP experiments implied that the 224-277 and 724-775 amino acid residues of Nat10 were the binding sites for Uqcr11 and Uqcrb mRNAs, respectively (Supplementary Fig. 9c). Nat10-Δ224-277 and Nat10-Δ724-775 had no inhibitory effect on Uqcr11 and Uqcrb mRNAs' nuclear export and protein expression, respectively (Supplementary Fig. 9d–f). Moreover, RNA pull-down experiment displayed that wild-type Uqcr11 and Uqcrb mRNAs could pull down Nat10, while the mutants of Uqcr11 and Uqcrb either lost or weakened their interaction with Nat10 (Supplementary Fig. 9g). This indicates that Nat10 binds specifically to Uqcr11 and Uqcrb mRNAs.

To provide further evidence that the regulation of Uqcr11 and Uqcrb by Nat10 is not dependent on ac4C modification, we conducted a mutation of the ac4C modification function in Nat10 (Nat10 G641E) according to the literature report[39] (Fig. 4d). Ac4C dot blot hybridization showed that overexpression of Nat10 increased the ac4C modification level of RNA in cardiomyocytes, while the ac4C modification after the mutation of Nat10 was abrogated (Fig. 4e). Meanwhile, Nat10 G641E was also able to increase the expression of Nat10 (Fig. 4f). Furthermore, the polysome profiling results showed that, although losing the function of ac4C modification, Nat10 G641E could still inhibit the expression of Uqcr11 and Uqcrb (Fig. 4g and

Supplementary Fig. 10). These results suggest that Nat10 non-classically reduces the translation of Uqcr11 and Uqcrb in an ac4C-independent manner.

To explore the effects of Uqcr11, Uqcrb, and Atp5j2 on cardiomyocyte proliferation and renewal, which were down-regulated during heart regeneration, we detected their effects on the cell cycle of cardiomyocytes. The results showed that the number of EdU, pH3 and Ki67-positive cardiomyocytes did not change after knocking down Atp5j2 (Fig. 4h and Supplementary Fig. 11). However, Uqcrb siRNA significantly increased the number of EdU and Ki67-positive cardiomyocytes, whereas it slightly increased the number of pH3-positive cardiomyocytes (Fig. 4h and Supplementary Fig. 11). Moreover, cardiomyocytes expressing EdU, pH3, and Ki67 were markedly increased after interfering with the expression of Uqcr11 (Fig. 4h and Supplementary Fig. 11). The above results indicate that Uqcr11 and Uqcrb negatively regulated by Nat10 inhibit cardiomyocyte proliferation. However, it is not clear yet whether Nat10 can regulate cardiomyocyte cycling and cardiac regeneration. Therefore, we next investigate the role and mechanism of Nat10 in cardiac regeneration.

### Nat10 regulates cardiomyocyte proliferation by targeting Uqcr11 and Uqcrb and mediating metabolic reprogramming

To further determine the role of Nat10 in heart regeneration, we used western blot to detect the expression of Nat10 in the heart tissue of mice subjected to AR and at different times after birth. The results showed that the expression of Nat10 in the heart tissues was up-regulated after AR, while down-regulated after birth (Fig. 5a, b). In order to monitor the expression changes of Nat10 in cardiomyocytes, we isolated cardiomyocytes from cardiac tissues of P1, P7, and adult mice and detected Nat10 expression. The results showed that the protein level of Nat10 in cardiomyocytes declined gradually with the increase in birth time (Fig. 5c). This illustrates that Nat10 has a critical regulatory role in cardiomyocyte cycling and heart regeneration.

Then, the EdU incorporation assay and cellular immuno-fluorescence technology were used to detect the regulatory effect of Nat10 on cardiomyocyte proliferation. The results showed that over-expression of Nat10 raised the positive rate of Ki67, EdU, and pH3 in cardiomyocytes (Fig. 5d). On the contrary, interfering with the expression of Nat10 reduced the percentage of Ki67, EdU, and pH3 (Fig. 5d and Supplementary Fig. 11). These results indicate that Nat10 could regulate the proliferation of cardiomyocytes. Moreover, we also detected the regulatory effect of Nat10 on HL1 proliferation. Similar to the results obtained from isolated cardiomyocytes from the heart, Nat10 overexpression promoted HL-1 proliferation, while knockdown of Nat10 inhibited HL-1 proliferation (Supplementary Fig. 12). Given that we have proven that Nat10 inhibits the non-classical translation of Uqcr11 and Uqcrb by affecting its distribution, we tested whether the target genes could rescue the regulatory effect of Nat10 on cardiomyocyte cell cycle entry. The outcomes of immunofluorescence staining showed that the reduction of EdU, Ki67, and pH3 positive cardiomyocytes occurred due to Nat10 knockdown, while the proliferation rate increased when Uqcr11 was knocked down in conjunction with Nat10 silencing in cardiomyocytes (Fig. 5e). Uqcrb could partially reverse the regulatory effect of interfering with Nat10 on the cardiomyocyte cell cycle, while silencing the expression of Uqcrb and Uqcr11 simultaneously blocked the inhibitory influence of Nat10 siRNA in cardiomyocyte proliferation to a greater extent (Fig. 5f). The above results indicate that Nat10 regulates cardiomyocyte proliferation by targeting Uqcr11 and Uqcrb, and may play a crucial role in cardiac regeneration and repair.

The conversion of metabolic mode from mitochondrial oxidative phosphorylation to glycolysis contributes to the activation of cardiomyocyte proliferation[7]. In the Ribo-seq data of cardiac regeneration, the RPFs and TE of mitochondrial complex components related to oxidative phosphorylation were mainly down-regulated, while the TE

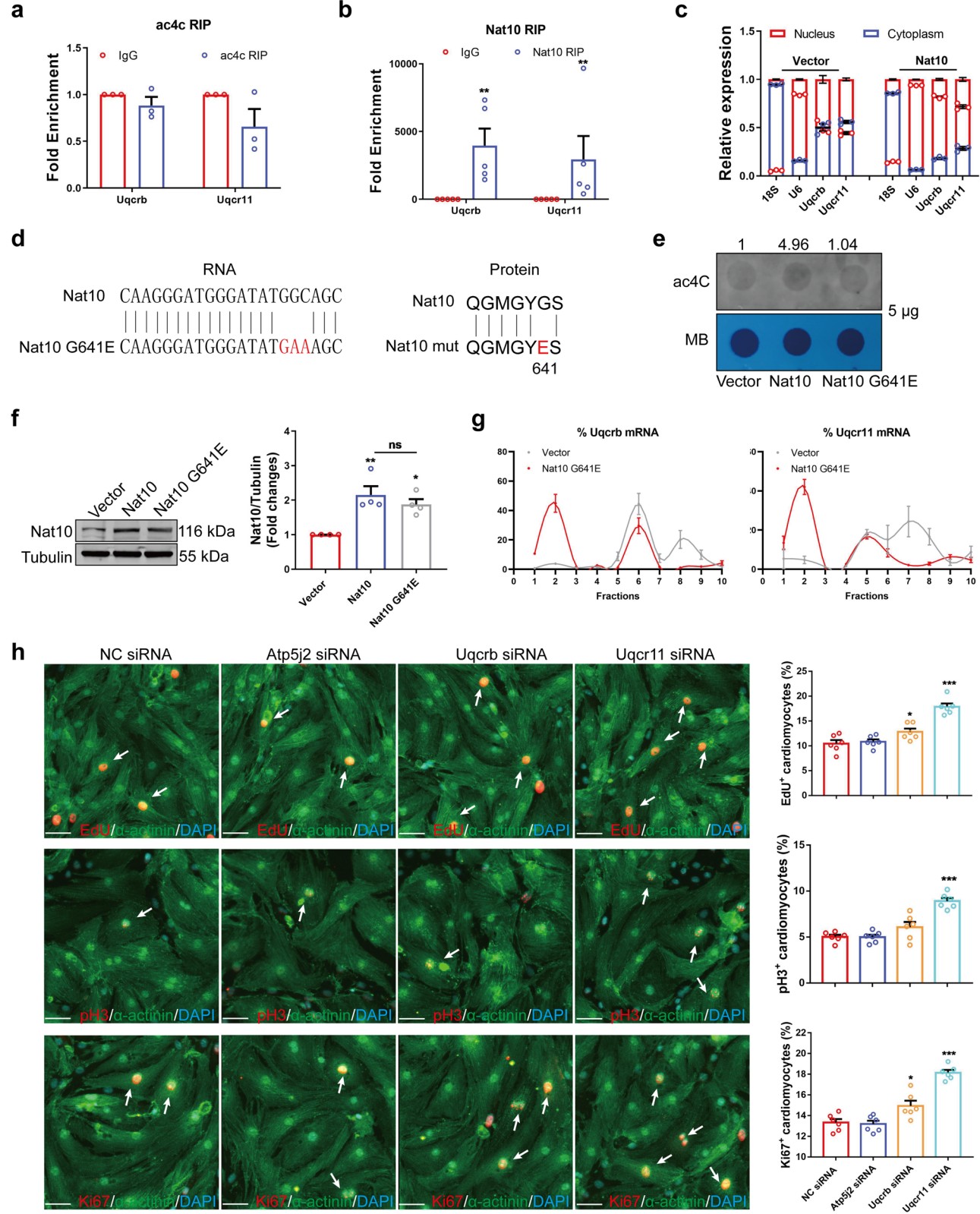

of glycolysis-related gene Pkm was up-regulated, indicating that the metabolic mode changes during cardiac regeneration (Supplementary Fig. 13). It is not clear whether Nat10 regulates cardiomyocyte proliferation by mediating metabolic reprogramming. As Uqcrb and Uqcr11 are components of mitochondrial complex III, we detected the effect of Nat10 on the activity of mitochondrial complex III and found that overexpression of Nat10 inhibited its activity, while knockdown of

Nat10 enhanced it (Fig. 5g). Additionally, the effects of Nat10 on cardiomyocyte metabolism were assessed by the seahorse mitochondrial stress test and glycolysis stress test. The results showed that the overexpression of Nat10 reduced the basic respiration, maximal respiration, and spare respiratory capacity, and increased the glycolytic capacity (Fig. 5h and i). Knockdown of Nat10 inhibited the glycolytic capacity, but silencing Uqcrb and Uqcr11 simultaneously

**Fig. 4 | The translation of Uqcr11 and Uqcrb mediating cardiomyocyte proliferation is regulated by Nat10 in an ac4C-independent manner. a** Ac4C RIP analysis of Uqcr11 and Uqcrb (*n* = 3 independent experiments). Two-tailed Student's *t* test with Welch's correction. **b** Nat10 RIP analysis of Uqcr11 and Uqcrb (*n* = 5 independent experiments). Two-tailed Mann–Whitney two-tailed test. **p** < 0.01 vs. IgG. **c** The distribution of Uqcr11 and Uqcrb between the nucleus and cytoplasm (*n* = 3 independent experiments). **d** The mutant site of Nat10. **e** The ac4C dot blot analysis in cardiomyocytes with WT or mut Nat10 overexpression. **f** Representative western blots and averaged data showing increased Nat10 in Nat10 and mutant Nat10 overexpressed cardiomyocytes compared with vector (*n* = 4 independent experiments). one-way ANOVA followed by Tukey's Multiple Comparison tests. **p** < 0.01 vs. Vector. **g** Polysome profiling analysis of Uqcrb and Uqcr11 in mutant Nat10 overexpressed cardiomyocytes (*n* = 4 independent experiments). **h** The effects of Uqcrb, Uqcr11 and Atp5j2 on cardiomyocyte proliferation analyzed by the detection of EdU, pH3 and Ki67 (*n* = 6 independent experiments). Scale bar: 50 μm. White arrows point the positive cardiomyocytes. One-way ANOVA followed by Dunnett's Multiple Comparison tests. *p < 0.05, ***p < 0.001 vs. NC siRNA. Data are presented as mean ± SEM. Source data are provided as a Source Data file. Exact *p* values are provided in the Source Data file.

reversed the inhibitory effect of Nat10 siRNA on glycolysis (Fig. 5j). These data indicate that Nat10 affects the metabolism of cardiomyocytes by regulating the expression of Uqcrb and Uqcr11, thus mediating the proliferation of cardiomyocytes.

## Nat10 is required for heart regeneration after injury

To further certify the regulatory effects of Nat10 on cardiac regeneration, we constructed cardiomyocyte-specific Nat10 knockdown mice (Fig. 6a). The mice were subjected to apex resection on the first day of birth. Twenty-eight days after the surgery, HE staining showed that the cardiomyocyte-specific knockdown of Nat10 inhibited cardiac regeneration (Fig. 6b). The mouse cardiac ultrasound results showed that knockdown of Nat10 weakened the heart's contractile function (Fig. 6c). The proliferative ability of cardiomyocyte was detected by tissue fluorescence staining 7 days after AR operation in mice. The results showed that the number of pH3- and Ki67-positive cardiomyocytes in Nat10 flox/+ mice significantly increased after AR, while specific knockdown of Nat10 in cardiomyocytes decreased the number of pH3- and Ki67-positive cardiomyocytes (Fig. 6d, e). Furthermore, the translation and expression of Uqcrb and Uqcr11 in Nat10 knockdown heart tissue was detected using polysome profiling and western blot. The results showed that the translation and expression of Uqcr11 and Uqcrb in Nat10 knockdown heart tissue were up-regulated compared with control (Fig. 6f, g and Supplementary Fig. 14a). The above results show that the cardiomyocyte-specific knockdown of Nat10 significantly inhibits the cell cycling of cardiomyocytes and cardiac repair and regeneration after injury by promoting the expression of Uqcr11 and Uqcrb.

## Cardiac-specific overexpression of Nat10 promotes heart repair after injury

In order to determine the potential of Nat10 overexpression in promoting cardiac regeneration, we constructed cardiomyocyte-specific Nat10 overexpression mice and subjected them to left anterior descending coronary artery ligation. The results of tissue immunofluorescence showed that specific overexpression of Nat10 in cardiomyocytes could significantly promote the expression of proliferation markers pH3 and Ki67 (Fig. 7a, b). Cardiac function was comparable between wild-type and Nat10 overexpression mice in the sham operation group. However, 4 weeks after MI, Nat10 overexpression could enhance the cardiac contractile function of MI mice and reduce the infarct area (Fig. 7c, d). In addition, we observed a significant decrease in the protein expression levels of Uqcrb and Uqcr11 in the heart tissue of Nat10 overexpression mice compared to wild-type mice (Fig. 7e, f and Supplementary Fig. 14b). These results show that Nat10 promotes the regeneration and repair of the injured heart.

## Nat10 as a potential target in heart regeneration in pigs and humans

Having observed the effects of our discoveries on mice, we proceeded to investigate the clinical relevance in large mammals and humans. The expression of Nat10 in heart tissues and isolated cardiomyocytes from P1 and P7 pigs was first analyzed. Consistent with the findings in mice, the protein level of Nat10 was decreased in P7 compared to P1 in

both heart tissues and cardiomyocytes (Fig. 8a, b). This implies that Nat10 also possesses the ability to regulate heart regeneration in large mammals. To reflect the importance and involvement of Uqcr11 and Uqcrb in heart regeneration in large mammals, we analyzed their translational regulation in P1 and P7 pig hearts. The results demonstrated the translational level of Uqcr11 and Uqcrb was notably higher in P7 hearts than in P1 hearts, emphasizing their importance in heart regeneration (Fig. 8c and Supplementary Fig. 14c). Furthermore, we examined the expression changes of Nat10 in cardiomyocytes isolated from human failing and non-failing hearts. Our data showed that the protein level of Nat10 declined in cardiomyocytes of failing hearts (Fig. 8d). Moreover, to evaluate the effects of Nat10 on human cardiomyocyte proliferation, we used cardiomyocytes derived from human embryonic stem cells (hESC-CMs) and treated them with Nat10 overexpression plasmid and Nat10 siRNA. As shown in Fig. 8e, overexpression of Nat10 promoted hESC-CM proliferation, whereas inhibition of Nat10 inhibited hESC-CM proliferation. The same conclusion was obtained with the AC16 human cardiomyocyte cell line (Fig. 8f). These findings further highlight the role of Nat10 in regulating human cardiomyocyte proliferation and heart repair. In addition, to ascertain the potential involvement of the Nat10/Uqcr11-Uqcrb axis in human cardiomyocytes, we further observed the regulation of Nat10 in Uqcr11 and Uqcrb translation in human cardiomyocytes. Data showed that Nat10 overexpression inhibited the translation of Uqcr11 and Uqcrb (Fig. 8g and Supplementary Fig. 14d). These data indicate that Nat10 also causes an enhanced proliferative impact on human cardiomyocytes by inhibiting Uqcr11 and Uqcrb expression, highlighting the translational regulation potential of Nat10 in the treatment of heart diseases.

## Remodelin impairs cardiomyocyte proliferative potential and cardiac repair

Remodelin has been proven to be an inhibitor of Nat10. We investigated the effects of Remodelin on cardiomyocyte proliferation and heart regeneration in mice following AR. We first determined the regulatory effect of Remodelin on cardiomyocyte cycling. The results showed that Remodelin had an inhibitory effect on the proliferation of cardiomyocytes isolated from mouse hearts, HL1 cells, and AC16 cells (Fig. 9a, b, Supplementary Fig. 15). In vivo, we performed AR on newborn mice and simultaneously administered Remodelin. The downregulation of Nat10 expression by Remodelin was confirmed using western blot analysis (Fig. 9c). Consistent with the cardiomyocyte-specific knockdown of Nat10, Remodelin weakened the heart function of mice, inhibited the regeneration and repair capacity after heart injury, as assessed by echocardiography and HE staining (Fig. 9d, e). Moreover, as expected, the increase in the number of pH3- and Ki67-positive cardiomyocytes triggered by AR injury was obviously inhibited by Remodelin (Fig. 9f, g). Polysome profiling and western blot further confirmed that Remodelin promoted the protein expression of Uqcr11 and Uqcrb (Fig. 9h, i, Supplementary Fig. 16). These data imply that Remodelin attenuates neonatal heart regeneration.

To further clarify the mechanism of Remodelin in regulating the expression of Uqcr11 and Uqcrb, we overexpressed wild-type Nat10 and Nat10 mutants (Nat10-G641E, Nat10-Δ224-277 and Nat10-Δ724-775) in

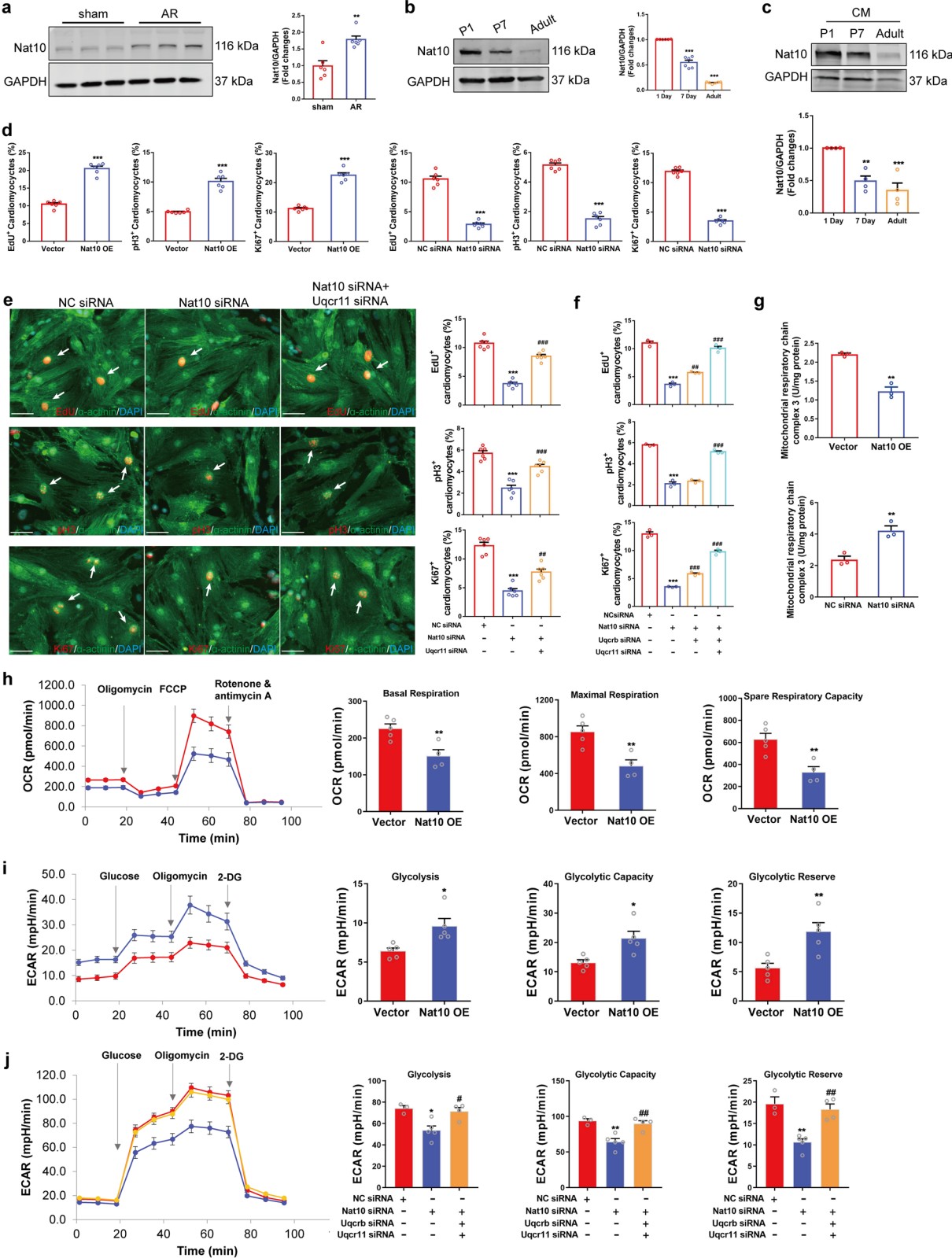

cardiomyocytes, and detected the effect of Remodelin on the expression of Uqcr11 and Uqcrb proteins, and the acetyltransferase active of Nat10 (Supplementary Fig. 17). The results indicate that Remodelin regulates the expression of Uqcr11 and Uqcrb by inhibiting the Nat10 expression, affecting the interaction of Nat10 with Uqcr11 and Uqcrb mRNA, rather than depending on the acetyltransferase activity of Nat10, thus regulating cardiomyocyte proliferation and cardiac repair.

## Nat10 is regulated by transcriptional factor Hes1

As the transcriptional level of Nat10 is up-regulated in the AR group compared to the sham group, therefore we searched for the vital transcription factors regulating Nat10 expression. Hes1 was predicted to be potential to bind to the promoter region of Nat10 by both PROMO and JASPAR coupled with UCSC Genome Browser (http://genome.ucsc. edu). And four binding sites were predicted in the Nat10 promoter

**Fig. 5 | Nat10 regulates cardiomyocyte proliferation by targeting Uqcr11 and Uqcrb. a** The expression of Nat10 in hearts of sham and AR mice ($n = 6$ independent mice). **$p < 0.01$ vs. sham group. **b** The expression of Nat10 in the heart tissues at different times points after birth ($n = 6$ independent mice). ***$p < 0.001$ vs. P1. **c** The expression of Nat10 in cardiomyocytes isolated from heart tissues at different times points after birth ($n = 4$ independent experiments from 8 mice per group). **$p < 0.01$, ***$p < 0.001$ vs. P1. **d** Immunofluorescence staining showing the expression of EdU, pH3 and Ki67 in cardiomyocytes with overexpression or knockdown of Nat10 ($n = 6$ independent experiments). ***$p < 0.001$ vs. Vector or NC siRNA. **e, f** The effects of Nat10 knockdown and rescued by Uqcr11/Uqcrb knockdown on cardiomyocyte cell cycle determined by immunofluorescence staining (**e**, $n = 6$ independent experiments; **f**, $n = 3$ independent experiments). ***$p < 0.001$ vs. NC siRNA, ##$p < 0.01$, ###$p < 0.001$ vs. Nat10 siRNA. White arrows point the region (Supplementary Fig. 18a). positive cardiomyocytes. **g** The activity of mitochondrial complex III in Nat10-upregulated and -downregulated cardiomyocytes ($n = 3$ independent experiments). **$p < 0.01$ vs. Vector or NC siRNA. **h** Mitochondrial stress test in cardiomyocytes (Vector, $n = 5$ independent experiments; Nat10 OE, $n = 4$ independent experiments). **$p < 0.01$ vs. Vector. **i** Glycolysis stress test in cardiomyocytes ($n = 5$ independent experiments). *$p < 0.05$, **$p < 0.01$ vs. Vector. **j** Nat10 regulated glycolysis targeting Uqcr11 and Uqcrb ($n = 3, 5, 4$ independent experiments per group). *$p < 0.05$, **$p < 0.01$ vs. NC siRNA, #$p < 0.05$, ##$p < 0.01$ vs. Nat10 siRNA. Data were analyzed using Mann–Whitney two-tailed U test **a**, one-way ANOVA followed by Tukey's Multiple Comparison tests **b, c, e, f, j**. Two-tailed Student's $t$ test with or without Welch's correction **d, g, h, i**. Data are presented as mean ± SEM. Scale bar: 50 μm **e**. Source data are provided as a Source Data file. Exact $p$ values are provided in the Source Data file.

region (Supplementary Fig. 18a). To confirm the regulatory effect of Hes1 on Nat10 in cardiomyocytes, we constructed luciferase reporter vectors containing the full-length promoter of Nat10. The dual-luciferase reporter assays revealed that overexpression of Hes1 led to a promotion of Nat10 promoter activity (Supplementary Fig. 18b). Sequential deletions of the Nat10 promoter, according to the presumptive binding sites, uncovered that the deletion of a Nat10 promoter containing fragment of 1500–1509 resulted in the disappearance of Hes1 overexpression activation of Nat10 transcriptional activity (Supplementary Fig. 18b). Subsequently, we detected the effect of Hes1 on Nat10 expression. The results showed that overexpression of Hes1 promoted, while Hes1 siRNA reduced the expression of Nat10 (Supplementary Fig. 18c). We also observed the expression of Hes1 in P1, P7, and AR hearts. The results showed that Hes1 is decreased in P7 heart compared to P1 heart, while increased in AR hearts compared to sham (Supplementary Fig. 18D and E). These results imply that the expression of Nat10 is regulated by its transcription factor, Hes1.

## Discussion

IHD is one of the leading causes of death which largely affects life quality worldwide. The identification of key targets for promoting endogenous cardiomyocyte proliferation and heart regeneration is of great significance in repairing the damaged heart. Currently, the targets regulating heart regeneration are identified mainly at the transcriptional level. However, translational profiling has not yet been revealed. To address this gap, we employed Ribo-seq and RNA-seq to profile the translatomics in heart regeneration and repair. Our findings indicate that the genes subject to translational regulation are primarily related to OXPHOS. Furthermore, it was clarified that Nat10 regulates the expression of Uqcrb and Uqcr11 through non-acetylation modification. Then, we identified Nat10 as a crucial epigenetic regulator that mediates the translation of OXPHOS-related genes Uqcr11 and Uqcrb in a non-classical manner, regulating cardiomyocyte cell cycle and cardiac regeneration and repair. Additionally, Nat10's expression level was decreased in the cardiomyocytes of failing hearts in patients. Nat10 also had the ability to regulate the proliferation of cardiomyocytes induced from human embryonic stem cells (hESC-CMs) and the translation of Uqcr11 and Uqcrb in human cardiomyocytes.

Translation regulation is a crucial process that all protein-coding genes undergo, which is associated with RNA translation, transport, localization, etc. Ribo-seq, as a novel technique, aids in identifying the translation regulation of various cellular processes. It has been applied to manifest the translational changes during the differentiation of forebrain neuron and plant immune induction, providing strong evidence for the overall translation reprogramming and the underlying mechanism[15,40]. Furthermore, translation regulation analysis has also been carried out on the heart tissue of patients with dilated cardiomyopathy (DCM) to demonstrate the extensive translation control of cardiac genes expressed in the DCM disease[16]. The use of Ribo-seq has enabled researchers to analyze the translatome changes in the process of cardiac hypertrophy[19]. Genome-wide changes in gene translation during the development of heart failure and cardiac fibrosis have also been studied[20,41]. However, translatomics in the process of heart regeneration has not been studied yet.

In this study, we conducted translatomics research on regenerative heart tissues to reveal the overall translation regulation. Our findings indicate that transcriptionally and translationally differential genes were devoid of consistency during heart regeneration. Of note, the genes involved in both transcriptional and translational regulation only account for a small portion whereas some genes exhibited completely opposite changes. This indicates that gene regulation during heart regeneration is complicated and precise. As Ribo-seq is intensely associated with protein synthesis and function[16], we adopted stringent conditions to analyze genes that undergo translational regulation without corresponding changes in RNA and ultimately lead to changes in expression. Data showed that terms of oxidative phosphorylation associated with cardiac energy metabolism, as well as protein digestion and absorption associated with extracellular matrix were enriched. Specifically, we focused on OXPHOS-related genes that were translationally regulated and found that the translation level of Uqcr11 and Uqcrb was reduced by Nat10. We also confirmed that Nat10 shifts the transition of OXPHOS to glycolysis by targeting Uqcr11 and Uqcrb, leading to metabolic reprogramming of cardiomyocytes. Extracellular matrix also plays critical roles in heart regeneration[9], which is altered in regenerative hearts as well.

Nat10, an RNA ac4C modification enzyme, is involved in a variety of cellular processes and has been linked to several diseases. A recent study has demonstrated that NAT10 enhances the translation efficiency of CEP170 by acetylating its mRNA, thereby promoting the proliferation of multiple myeloma cells[24]. NAT10 also promotes the osteogenic differentiation of bone marrow mesenchymal stem cells by increasing the level of ac4C modification of RUNX2 mRNA[42]. In colorectal cancer cells, NAT10 was found to enhance the stability of KIF23 mRNA by up-regulating its ac4C modification. This subsequently activated Wnt/β-Catenin pathway, inhibiting cell apoptosis and enhancing cell proliferation[21]. Nat10 also plays an important role in regulating cell proliferation as a protein acetylase. It acetylates microtubule protein to maintain its stability which is crucial for cell cycle control during mouse oocyte meiosis[43]. Consistently, the specific elimination of Nat10 in germ cells severely inhibits the entry of meiosis[44]. In addition, Nat10 regulates the fate of mitotic cells by acetylating Eg5 at K771. Decreasing Nat10 reduces the stability of Eg5, leading to the formation of unipolar and asymmetric spindles and affecting mitosis[45]. Of note, Nat10 can regulate the same gene in different ways. In gastric cancer progression, NAT10 modifies MDM2 mRNA with ac4C acetylation to stabilize MDM2 mRNA, resulting in its up-regulation and down-regulation of p53[46]. In colorectal cancer cells, NAT10 acetylates p53 at K120 and promotes the degradation of MDM2 by binding with MDM2 protein, which mediates the regulation of the cell cycle and apoptosis[47].

The impact of Nat10 on cancer cell proliferation has been established through its regulation of protein and RNA acetylation. In this

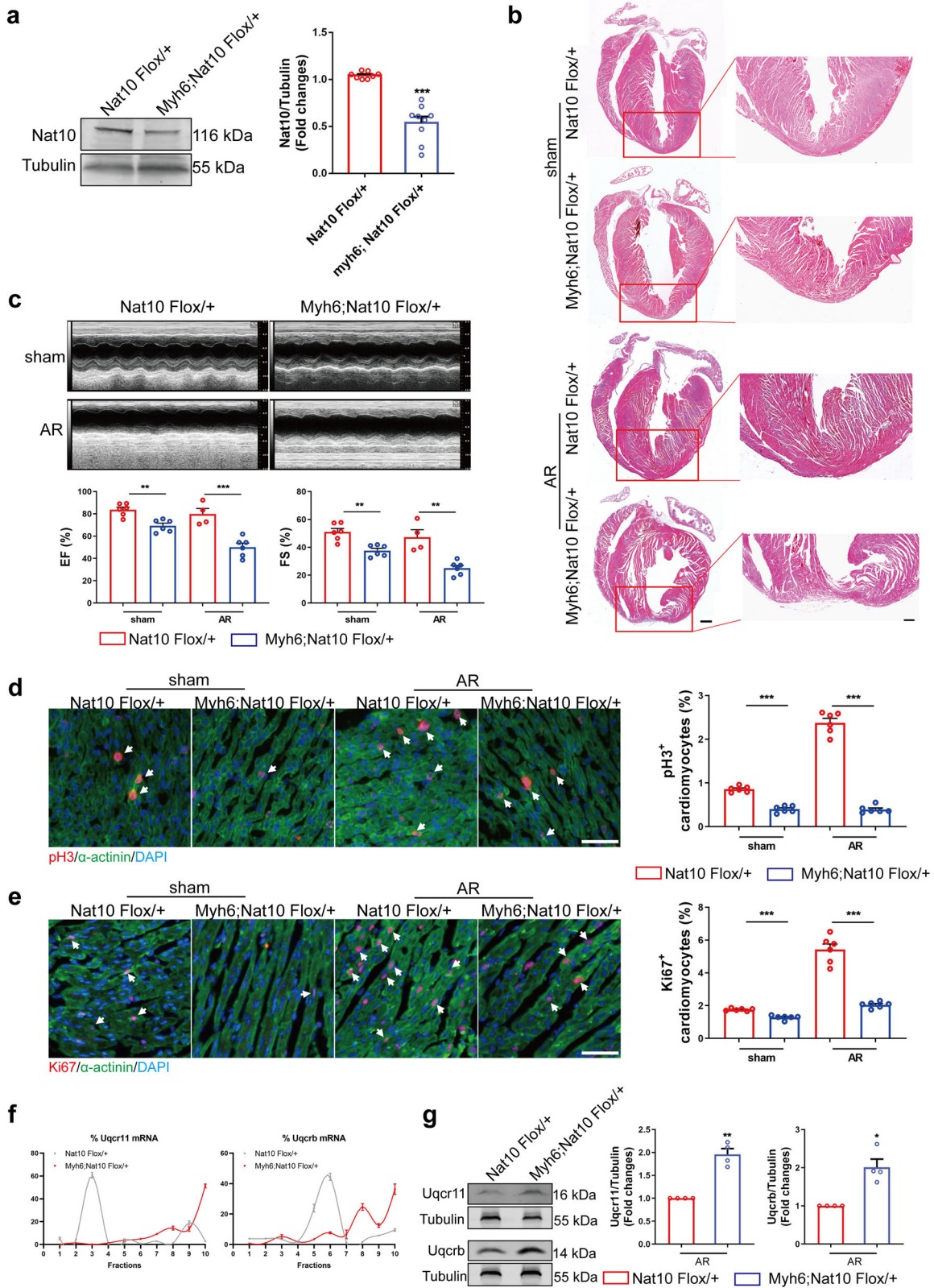

study, we verified the regulatory effect of Nat10 on cardiomyocyte proliferation and found that it achieved this effect by reducing the quantity of translatable Uqcrb and Uqcr11 mRNA in ribosomes. Interestingly, unlike previous studies, Nat10 regulated the translation of Uqcrb and Uqcr11 in an ac4C-independent manner. The non-ac4C-dependent characteristics of Nat10 in regulating translation were further corroborated by employing mutations in the ac4C modification active site of Nat10, as reported in previous literature[39]. Our study revealed that Nat10 decreases the expression of Uqcr11 and Uqcrb in a non-classical model of translational regulation that does not affect translation initiation and elongation like traditional translational regulation. Instead, it affects translation and expression by binding to the mRNA of Uqcrb and Uqcr11, blocking their nuclear export and reducing the number of translatable mRNAs. This finding is also consistent

**Fig. 6 | The regenerative ability of cardiomyocyte-specific Nat10 knockdown mice. a** Representative western blots and statistic data showing Nat10 expression in the hearts from adult Nat10 knockdown mice (Myh6;Nat10 Flox/+) and control mice (Nat10 Flox/+) (Nat10 Flox/+, $n = 8$ mice; Myh6;Nat10 Flox/+, $n = 9$ mice). $***p < 0.001$ vs. Nat10 Flox/+. **b** Morphology of Nat10 Flox/+ and Myh6;Nat10 Flox/+ mouse hearts at 28 days after sham or AR surgery analyzed by HE staining ($n = 3$ mice). Scale bar: 500 μm (left) and 200 μm (right). **c** Echocardiographic analysis of Nat10 Flox/+ and Myh6;Nat10 Flox/+ mice ($n = 6, 6, 4, 6$ mice per group). $**p < 0.01$, $***p < 0.001$ as indicated. **d** Representative heart tissue sections stained with pH3 at P7 after AR, and the quantification of the percentage of pH3$^+$ cardiomyocytes ($n = 6$ mice). $***p < 0.001$ as indicated. White arrows point the positive cardiomyocytes. **e** Representative heart tissue sections stained for Ki67 at P7 after AR, and the quantification of the percentage of Ki67$^+$ cardiomyocytes ($n = 6$ mice). $***p < 0.001$ as indicated. White arrows point the positive cardiomyocytes. **f** Polysome profiling analysis of Uqcrb and Uqcr11 ($n = 3$ mice). **g** Western blot analysis of Uqcr11 and Uqcrb in Nat10 Flox/+ and Myh6;Nat10 Flox/+ mouse hearts ($n = 4$ mice). $*p < 0.05$, $**p < 0.01$ vs. Nat10 Flox/+. Two-tailed Student's $t$ test with Welch's correction **a**, **e**, **g**. Two-tailed Student's $t$ test **c**–**e**. Data are presented as mean ± SEM. Scale bar: 50 μm **d**, **e**. Source data are provided as a Source Data file. Exact $p$ values are provided in the Source Data file.

with the sequencing data, as the RNA levels of Uqcrb and Uqcr11 have not changed, while the RPFs have decreased. Additionally, we also confirmed Nat10's pro-proliferative effect on human cardiomyocytes. These findings provide a theoretical basis for the clinical transformation of Nat10 in heart repair.

We also proved that Remodelin impairs cardiomyocyte proliferation and cardiac repair. Remodelin has been employed as capable of inhibiting the activity of Nat10's lysine acetyltransferasein many studies[24,48–50]. The evidence for interaction of Remodelin with the Nat10 acetyltransferase active site is limited so far[51]. Recent studies reported that Remodelin was able to inhibit the expression level of Nat10[21,39], suggesting Remodelin at least as an inhibitor of Nat10 expression. Consistent with these studies, our research also found that Remodelin reduced the level of Nat10 protein in cardiomyocytes. To better clarify the mechanism by which Remodelin regulates the expression of Uqcr11 and Uqcrb, we constructed Nat10 plasmids including wild-type Nat10, the acetylation-defective Nat10 mutant (Nat10-G641E), and the mutants Nat10 that loses their ability to bind with Uqcr11 (Nat10-△224-277) or Uqcrb (Nat10-△724-775) mRNAs, and demonstrated that Remodelin regulates Uqcr11 and Uqcrb expression by inhibiting the expression of Nat10, thus affecting the interaction of Nat10 with Uqcr11 and Uqcrb mRNAs, rather than depending on the acetyltransferase activity of Nat10.

Study limitations: Our research demonstrated the translational regulation of regenerative heart tissues at a general level, and we were unable to accurately describe the changes in translational regulation of various cells. To gain a more detailed understanding of the role of different cell types in heart regeneration, it is necessary to study the changes in translational regulation of various cells in regenerative heart tissue. Furthermore, as previous study has reported that the proliferation of cardiomyocytes peaks on the seventh day after injury[52], we obtained cardiac tissue six days after AR injury for translation regulation study to maximize the identification of the genes regulating this process. As multiple processes occur during heart regeneration, numerous genes might be expressed differentially at different time points. However, we only selected one time point for research, which could not fully display gene remodeling in the process of heart regeneration. Therefore, selecting multiple timepoints would provide a more complete understanding of gene remodeling during heart regeneration.

In summary, we conducted a comprehensive analysis of the translatomics landscape in cardiac regeneration, revealing the crucial role of Nat10 in the translational control of genes related to oxidative phosphorylation and cardiomyocyte proliferation. Mechanistic investigations revealed that Nat10 promoted cardiomyocyte proliferation by reprogramming the metabolic mode through decreasing the nuclear export and subsequent translation of mitochondrial components genes Uqcrb and Uqcr11. Transcriptional factor Hes1 acts as the upstream regulator of Nat10. Our findings indicate that targeting Nat10 could represent a promising therapeutic strategy for enhancing cardiomyocyte proliferation and facilitating heart repair following injury.

## Methods

### Ethics statement

The animal studies, including those on mice and pigs, were approved by the Ethics Committee of the Harbin Medical University. All procedures comply with the Guide for the Care and Use of Laboratory Animals published by the National Institutes of Health. Collection of human samples (3 female human failing and 3 female non-failing hearts) complied with the guidelines of the Ethics Committee of the Harbin Medical University, and written informed consent was obtained from all participants. We obtained human failing heart tissue samples from patients with cardiac valvular disease or atrial fibrillation. The non-failing heart tissue samples were obtained from patients with aortic dissection. Cardiomyocytes isolated from heart tissue were used for further analysis using western blot. All procedures were carried out in accordance with the Declaration of Helsinki.

### Animal experiments

The neonatal mice aged 1–2 days and male/female C57BL/6 mice aged 8–10 weeks were purchased from the Laboratory Animal Center of the Second Affiliated Hospital of Harbin Medical University (Harbin, China). Cardiomyocytes were isolated from neonatal mice and the adult mice were crossed to give birth to neonatal mice for AR. Nat10 cardiomyocyte-specific knockout mice on C57BL/6 background constructed with CRISPR/Cas9 were purchased from Cyagen (Suzhou, China), The Nat10 knockin transgenic mice on C57BL/6 background were purchased from Cyagen (Suzhou, China). The Myh6 promoter was applied to induce cardiomyocyte-specific expression of Cre-recombinase. The genotyping primers for Nat10 KO are forward primer CACAGAGGAGTGCCTCAGTTCTAG and reverse primer AAAGATGGGAAACAGGGGCAG; for Nat10 KI are forward primer CACTTGCTCTCCCAAAGTCGCTC and reverse primer ATACTCCGAGGCGGATCACAA; for Myh6-Cre are forward primer -GAAATGACAGACAGATCCCTCCTATC and reverse primer CGACGATGAAGCATGTTTAGCTG.

The apex resection (AR) was performed in P1 knockout mice and myocardial infarction experiments were performed in adult knockin male mice (8–10 weeks) as previously reported[53,54]. Briefly, for neonatal mice, lateral thoracotomy at the fourth intercostal space was performed by blunt dissection of the intercostal muscles following skin incision. Iridectomy scissors were used to resect the apex of P1 hearts. Then their thoracic wall incisions were sutured with 7–0 nonabsorbable silk suture, and the skin wounds were closed by using skin adhesive. For adult mice, they were anesthesia and received tracheal intubation. Then, the left anterior descending coronary artery was ligated with 7–0 prolene suture to construct MI model. Sham-operated mice underwent the same procedure without apical resection or ligation. The adult mice were anesthetized by intraperitoneal injection of a mixture of xylazine (5 mg/kg) and ketamine (100 mg/kg) and then euthanized by cervical dislocation. The P1 mice were treated with Remodelin after AR daily by intraperitoneal injection for 7 consecutive days at 25 mg/kg/day. Remodelin (Selleck, S7641) was dissolved in a solution of 5% DMSO, 40% PEG300, 5% Tween80, and 50% ddH$_2$O. The solution alone was given to mice for the vehicle group.

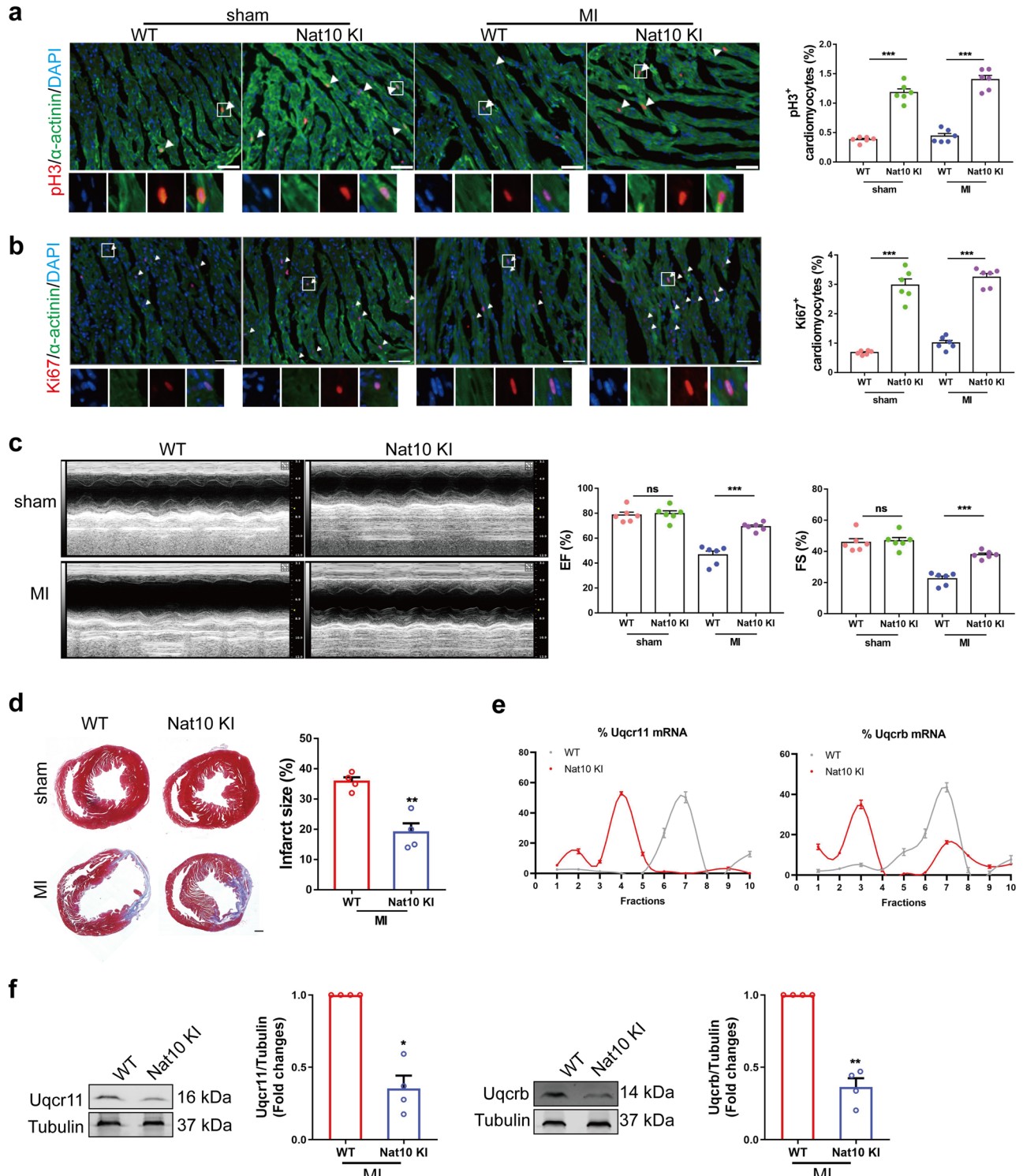

**Fig. 7 | Nat10 promotes cardiomyocyte proliferation and improves cardiac functions in adult mice with MI. a**, **b** Cardiac sections from hearts stained with pH3 and Ki67 at 7 days after MI. Arrows, pH3/Ki67-positive signal (*n* = 6 mice). ***$p < 0.001$ as indicated. White arrows point the positive cardiomyocytes. **c** Echocardiography was used to observe cardiac function of adult mice at 28 days after MI (*n* = 6 mice). ***$p < 0.001$ as indicated. **d** The fibrosis of adult mice after MI for one month was studied using Masson staining (*n* = 4 mice). **$p < 0.01$ vs. WT.

**e** Polysome profiling analysis of Uqcrb and Uqcr11 (*n* = 3 mice). **f** Representative western blots and averaged data showing downregulated Uqcr11 and Uqcrb in the hearts from WT and cardiomyocyte-specific Nat10 overexpression mouse hearts (*n* = 4 mice). *$p < 0.05$, **$p < 0.01$ vs. WT. Two-tailed Student's *t* test with (**a**, **b**, **f**, Uqcrb) or without **a**–**d** Welch's correction. Data are presented as mean ± SEM. Scale bar: 50 μm **a**, **b**, 500 μm **d**. Source data are provided as a Source Data file. Exact *p* values are provided in the Source Data file.

The postnatal day 1 (P1) and day 7 (P7) male bama miniature pigs were purchased from Harbin Veterinary Research Institute, Chinese Academy of Agricultural Sciences. The hearts of P1 and P7 pigs, which did not undergo any damage or experiments prior to harvesting, were harvested for western blot and polysome profiling analysis. Animals were housed in ventilated cages (mice) and rooms (pigs) under a temperature (20–24 °C) and humidity (40–60%) with a 12-h light/dark cycle and were fed with commercial formula feed

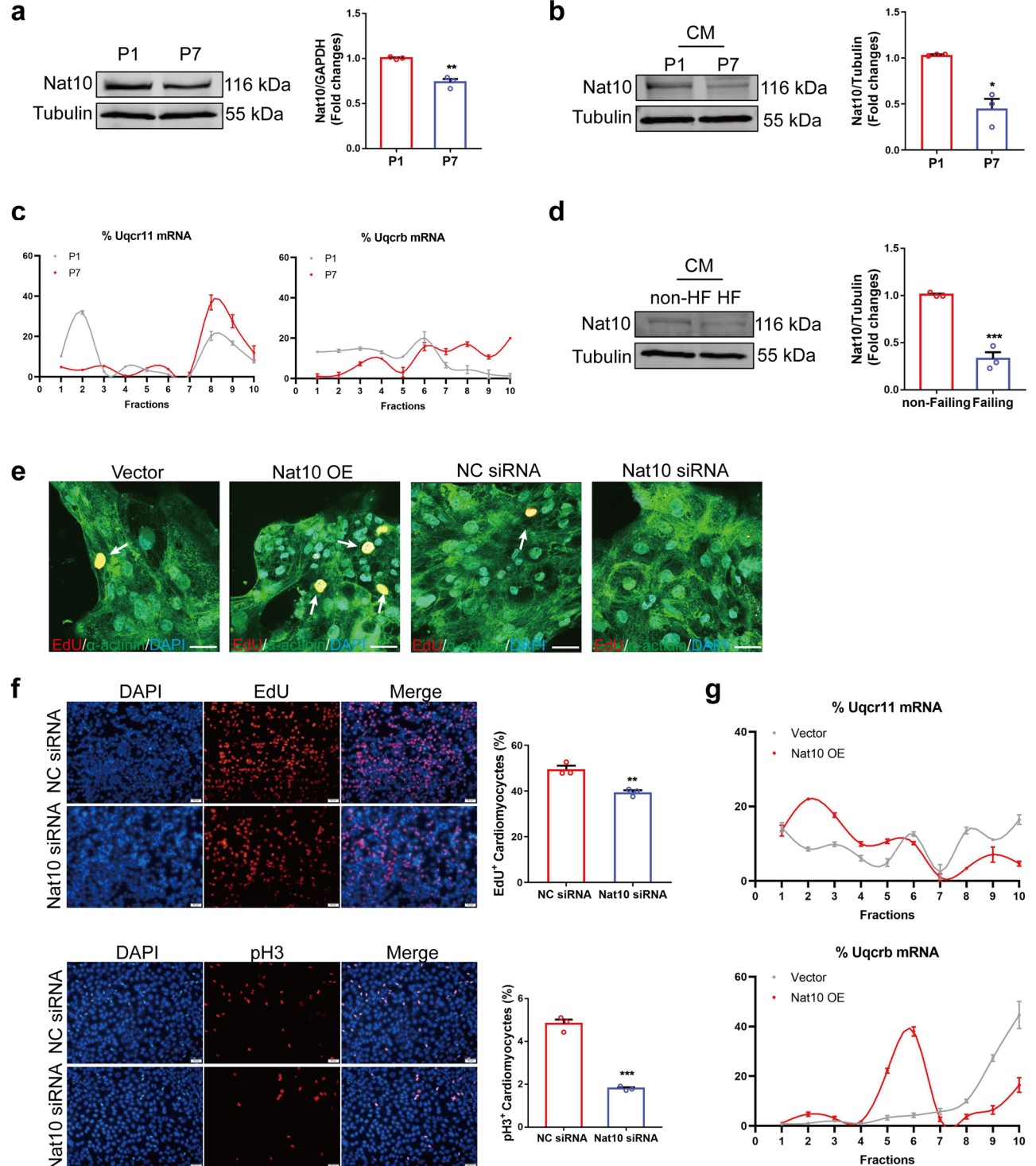

**Fig. 8 | Nat10 plays crucial role in heart regeneration in pigs and humans. a** The expression of Nat10 in male pig heart tissues after birth ($n = 3$ pigs). **$p < 0.01$ vs. P1. **b** The expression of Nat10 in cardiomyocytes isolated from male pig heart tissues ($n = 3$ pigs). *$p < 0.05$ vs. P1. **c** The translational regulation of Uqcrb and Uqcr11 in P1 and P7 male pig heart tissues analyzed by polysome profiling ($n = 3$ pigs). **d** The expression of Nat10 in cardiomyocytes isolated from female human failing and non-failing hearts ($n = 3$ hearts). ***$p < 0.001$ vs. non-HF. **e** Immunofluorescence staining showing the expression of EdU in cardiomyocytes derived from human ESCs with overexpression or knockdown of Nat10 ($n = 3$ independent experiments).

White arrows point the positive cardiomyocytes. **f** The effects of Nat10 knockdown on AC16 cardiomyocyte cell cycle determined by immunofluorescence staining ($n = 3$ independent experiments). **$p < 0.01$, ***$p < 0.001$ vs. NC siRNA. **g** Polysome profiling analysis of Uqcrb and Uqcr11 in Nat10-overexpressed human cardiomyocytes ($n = 3$ independent experiments). Two-tailed Student's $t$ test with **b**, **f** or without Welch's correction **a**, **d**. Data are presented as mean ± SEM. Scale bar: 50 μm **e**. Source data are provided as a Source Data file. Exact $p$ values are provided in the Source Data file.

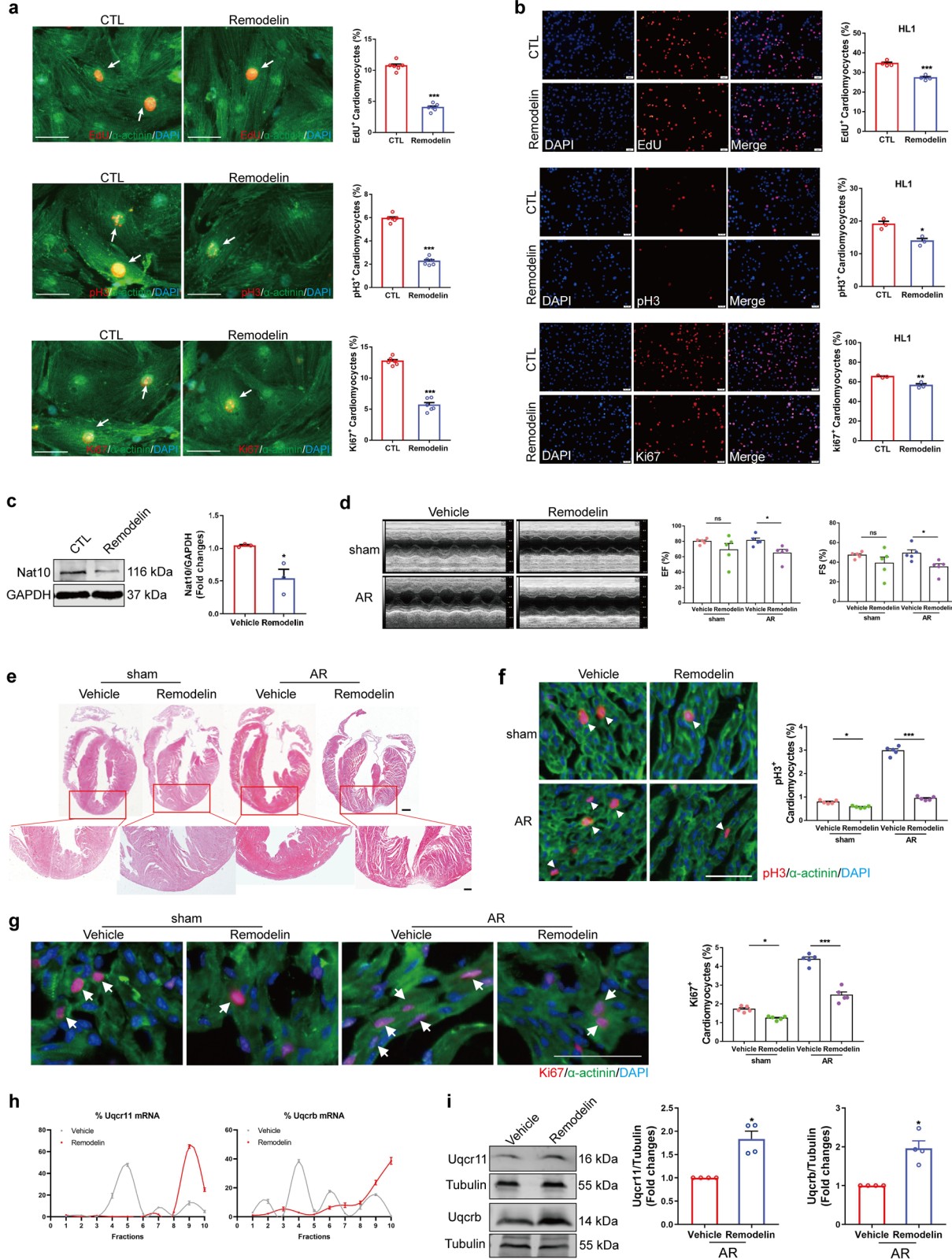

satisfying the standard mice and pig nutritional requirements and water ad libitum.

## Ribo-seq and RNA-seq analyses

The hearts were harvested 6 days after AR. The injury border zone which was defined as the lower third of the left ventricular myocardium from the level below the auricular appendix to the incision level was collective. One-third of the collected cardiac tissues were used for RNA-seq analysis and two-thirds was used for Ribo-seq analysis. For Ribo-seq analysis, the tissues were lysed using lysis solution (20 mM Tris-HCl, pH = 7.5, 150 mM NaCl, and 5 mM $MgCl_2$, 1% Triton X-100, 1 mM DTT, 10 U/mL DNase I, and 0.1 mg/mL cycloheximide, RNase-free $H_2O$) on ice for 10 min. The lysate was centrifugated at 20,000 g, 4 °C for 10 min.

**Fig. 9 | The effects of Remodelin on cardiomyocyte proliferation and heart regeneration. a** Remodelin inhibited the proliferation of cardiomyocytes isolated from P1 mouse hearts determined by pH3, EdU and Ki67 staining ($n = 6$ independent experiments). ***$p < 0.001$ vs. CTL. White arrows point the positive cardiomyocytes. **b** Remodelin inhibited the proliferation of HL1 cells determined by pH3 ($n = 3$ independent experiments), EdU ($n = 4$ independent experiments) and Ki67 ($n = 3$ independent experiments) staining. *$p < 0.05$, **$p < 0.01$, ***$p < 0.001$ vs. CTL. **c** Representative western blots and statistics showing downregulated Nat10 in the hearts from Remodelin treated or untreated mouse hearts ($n = 3$ mice). *$p < 0.05$ vs. Vehicle. **d** Cardiac function by echocardiography one month after AR ($n = 5$ mice).

*$p < 0.05$ as indicated. **e** HE staining at P28 after being treated with AR in P1 hearts ($n = 3$ mice). Scale bar: 500 μm (up) and 200 μm (down). **f, g** Heart sections stained with pH3, Ki67 and the corresponding statistics at 7 days after AR ($n = 5$ mice). *$p < 0.05$, ***$p < 0.001$ as indicated. White arrows point the positive cardiomyocytes. **h** Polysome profiling analysis of Uqcrb and Uqcr11 ($n = 3$ mice). **i** Western blot analysis of Uqcr11 and Uqcrb ($n = 4$ mice). *$p < 0.05$ vs. Vehicle. Two-tailed Student's $t$ test **a**–**c**, Two-tailed Student's $t$ test with Welch's correction **d, i**, one-way ANOVA followed by Tukey's Multiple Comparison tests **f, g**. Data are presented as mean ± SEM. Scale bar: 50 μm **a, b, f, g**. Source data are provided as a Source Data file. Exact $p$ values are provided in the Source Data file.

The Ribo sequencing service and the RNA sequencing service were provided by ShuPu (Shanghai) BIOTECHNOLOGY LLC using the Illumina sequencer according to a previous report[13,14]. For Ribo-seq, the lysate was digested using RNase to remove RNA that is not protected by ribosomes. High-density sucrose centrifugation was performed to obtain a single ribosome followed by the extraction and purification of RPFs. cDNA library was then constructed and sequenced after quality control. Products. Products between 146 and 154 bps were subjected to the purification and recovery (corresponding to 26–34 nt mRNA covered by ribosomes). 51 cycles were sequenced on the Illumina NextSeq 500 sequencer according to the supplier's instructions. Solexa pipeline version 1.8 (Off-Line Base Caller software, version 1.8) software was used for image processing and base recognition. FastQC software was applied to evaluate the sequencing quality of reads. Of note, RNA sequencing merely used R1 single-ended part (150 bp) to be consistent with Ribosomal sequencing. The software cutadapt was used to remove 3' and 5' joints and to filter reads with lengths less than 22 bp and more than 34 bp. Bowtie software was used to excluder rRNA/tRNA/snRNA/snoRNA and STAR software was used to compare with the reference genome. Unique mapped reads were used for All downstream analyses in this analysis. RPF QC was drawn directly using RiboTish software. Using Plastid software to estimate the translation abundance of gene CDS region and calculate RPKM with reference to the annotation information of ENSEMBL official database. Use edgeR package of R software to calculate the fold change, P value, and Q value of RPF. Accordingly, differential genes between 2 groups were screened. RiboDiff software was used to calculate TE's Fold change, P value, and Q value. Accordingly, genes with different TE were screened. For RNA-seq, total RNA samples were enriched by oligo dT (rRNA removal), and then KAPA Stranded RNA-Seq Library Prep Kit (Illumina) was selected to construct the library. In the library construction process, the double-stranded cDNA was synthesized by dUTP method combined with the subsequent high-fidelity PCR polymerase, so that the final RNA library was strand-specific. The quality of the constructed library was identified by Agilent 2100 Bioanalyzer, and the library was quantified by qPCR and sequenced with Mumina NovaSeq 6000 sequencer.

## Cells culture, transfection and treatment

Cardiomyocytes were isolated from neonatal mice by digesting the heart tissues with trypsin and then purified according to cell adhesion. The cells were cultured with medium containing 10% FBS and 1% penicillin-streptomycin in humidified incubator at 37 °C and 5% $CO_2$. Cells were transfected with Nat10 overexpression plasmid, Nat10 mut plasmids, and vector using Lipofectamine 2000 (Invitrogen, Carlsbad, CA, USA), and transfected with Nat10 siRNA (1: GCAUUCGGGUAUUCC AAUATT, UAUUGGAAUACCCGAAUGCTT; 2: CCCUCAGUCCUGUGGU GUTT, AACACCACAGGACUGAGGGTT), Uqcrb siRNA (1: GCAAGUGG CUGGAUGGUUUTT, AAACCAUCCAGCCACUUGCTT; 2: CCUAAGGA UCAGUGGACAATT, UUGUCCACUGAUCCUUAGGTT), Uqcr11 siRNA (1: AGUUUAAGAAGGACGAUUATT, UAAUCGUCCUUCUUAAACUTT; 2: ACAUCAACGGCAAGUUUAATT, UUAAACUUGCCGUUGAUGUTT), ATP5j2 siRNA (1: GUACAUCAACGUUCGGAAATT, UUUCCGAACGUUGA UGUACTT; 2: CUGCAUUUCUUACAAGGAATT, UUCCUUGUAAGA

AAUGCAGTT), and NC siRNA (UUCUCCGAACGUGUCACGUTT, ACG UGACACGUUCGGAGAATT) using RNAiMAX reagent (Invitrogen, Carlsbad, CA, USA) according to the manufacturers' instructions. The siRNAs were purchased from GenePharma Co., Ltd. (Shanghai, China), and the plasmids were purchased from Genechem Co., LTD. (Shanghai, China). The full length of Nat10, Hes1 and mutant Nat10 sequences were cloned into corresponding vectors to obtain overexpressed plasmids. Cells were treated with Remodelin at 2.5 μM or DMSO for 24 h. AC16 (human, ATCC, BFN60808678) and HL-1 (mouse, ATCC, BFN60808678) were obtained from BLUEFBIO life science. The cell lines were tested for mycoplasma using the Mycoplasma Detection Kit (Thermo Fisher Scientific), and were confirmed through short tandem repeat DNA profiling used within six months for testing.

The hESCs were obtained from the National Stem Cell Resource Center, and were maintained in an E8 medium (CA1001500, CELLAPY) at 37 °C and 5% $CO_2$. For monolayer cardiomyocyte differentiation, 70–80% confluent hESCs were cultured with differentiation basal medium comprising RPMI1640 medium (C11875500BT, Thermo Fisher Scientific) and B27 minus insulin (A1895601, Thermo Fisher Scientific). The hESCs were incubated in differentiation basal medium added with CHIR-99021 (HY-10182, MCE) for 1 day and Wnt-C59 (S7037, Selleck Chemicals) for 2 days, and the cells were subsequently fed every 1–2 days in RPMI1640 basal medium containing B27 (17504044, Thermo Fisher Scientific). Beating cells were observed at day 8–9 after differentiation.

## Immunofluorescence

Cells were fixed with 4% fixative solution (P1110, Solar IR) for 15 min at room temperature and then penetrated with 0.4% Triton-X100 (dissolved in 5 mg/mL BSA) for 90 min at room temperature. The cells were then blocked with goat serum (AR0009, Boster) for 30 min at 37 °C and incubated with antibodies to α-actinin (GTX29465, Gene Tex, 1:400), phosphoHistone H3 (pH3; #06–570, Millipore, Billerica, MA, USA, 1:400), Ki67(ab15580, Abcam, Cambridge, UK, 1:400) overnight at 4 °C. The secondary antibodies Alexa Fluor 488 (ab150113, Abcam, 1:400) and Alexa Fluor 594 (ab150080, Abcam, 1:400) were incubated with cells for 60 min at room temperature in the dark. After mounting and nucleic staining with DAPI (C0065; Solarbio, Beijing, China) for 15 min at room temperature, images were captured using confocal laser scanning microscope (FV300; Olympus, Japan) and analyzed with ImageJ software.

To detect cell proliferation, cells were incubated with 5-ethyl-2 '-deoxyuridine (EdU) for 12 h and then fixed, penetrated, and blocked as described above. Cardiomyocytes were counter-stained with α-actinin (GTX29465, Gene Tex, 1:400). The dye solution was prepared with EdU Apollo567 in Vitro Kit (Ribobio, Guangzhou, China) according to the instructions, and the cells were incubated in the dark for 30 min at room temperature. The nucleus is labeled with DAPI.

## Quantitative real-time PCR

Total RNAs were extracted from cells or heart tissues using TRIzol reagent (Invitrogen, Carlsbad, CA, USA) according to the manufacturer's guidelines and the concentration was determined with Nanodrop 2000 (Thermo ScientificTM, USA). SYBR Green PCR Master

Mix (Applied Biosystems, Foster City, CA) was used to amplify and determine cDNA from High Capacity cDNA Reverse Transcription Kit (Applied Biosystems, Foster City, CA) according to the manufacturer's instructions. Murine 18 S was used as loading control.

## Western blot

Proteins were extracted from cells or tissues using lysis. Protein concentration was measured using the Enhanced BCA Protein Assay Kit (P0010, Beyotime Biotechnology). The samples were resolved by SDS-PAGE and the target antigens were detected using following antibodies (Uqcrb (10756-1-AP, Proteintech, 1:1000), Uqcr11 (YN4606, Immunoway, 1:1000), ATP5j2 (68128-1-Ig, Proteintech, 1:1000), Nat10 (13365-1-AP, Proteintech, 1:1000), Hes1 (CY5649, Abways, 1:500), GAPDH (AB0037, Abways, 1:1000), and Tubulin (abs830032, Absin, 1:1000)). The images were captured using Odyssey system (LI-COR Biosciences, Lincoln, NE, USA).

## Fluorescence in situ hybridization (FISH)

The FISH probes of Uqcr11 and Uqcrb were designed and synthesized by GenePharma Co., Ltd. (Shanghai, China). FISH assays were performed according to the manufacturer's protocol. Briefly,

## RNA immunocoprecipitation (RIP)

Control and Nat10 overexpressing cell samples were collected and lysed with radioimmunoprecipitation (RIP) lysis buffer. A total of 50 μL magnetic beads were washed three times with RIP cleaning buffer and incubated with Nat10 antibody (13365-1-AP, Proteintech, 5 μg) at 4 °C overnight. Magnetic beads incubated with antibody were washed three times with RIP cleaning buffer and incubated with lysed samples at 4 °C overnight. Magnetic beads were washed six times with RIP cleaning buffer and eluted with 300 μL eluant buffer at 50 °C for 30 min. RNAs were extracted by TRIzol and analyzed by qRT-PCR.

## ac4C dot blot

Total RNAs from treated cardiomyocytes according to experience needs were extracted and denatured at 95 °C for 3 min. 5 μg RNAs were loaded on nylon membrane (YA1760, Solarbio) which was cross-linked for 40 min using 254 nm UV and blocked with 5% nonfat milk in 0.1% Tween 20 PBS for 1 h at room temperature. After immersion with methylene blue for 3 min, an anti-ac4C antibody (ab252215, Abcam, USA, 1:500) was incubated with the membrane at 4 °C overnight. The following steps were consistent with western bolt.

## Polysome profiling analysis

Cardiomyocytes transfected with Nat10 overexpression plasmid and mutant Nat10 overexpression plasmid were exposed to cycloheximide (CHX; 200 μg/mL) for 15 min and harvested. Cells were lysed with lysis buffer containing 20 mM Tris, pH 7.4, 15 mM $MgCl_2$, 200 mM KCl, 1% Triton X-100, 200 μg/mL CHX, 1 mM dithiothreitol, and 40 U/mL RNasin. The lysate was loaded to 10–50% linear sucrose gradient and centrifuged at 247606 g for 3 h. Different sucrose gradients after centrifugation were evenly divided into ten fractions and RNAs were extracted from each fraction and qRT-PCR was performed to examine the distribution of targets in different ribosome fractions.

## Seahorse analyses

The Agilent Seahorse XFe24 cellular energy metabolism analysis system (Agilent Technologies) was used to evaluate the energy metabolism of cardiomyocytes. The mitochondrial respiration of cardiomyocytes was analyzed using Seahorse XF Cell Mito Stress Test Kit (103015-100, Agilent Technologies, CA, USA) by measuring the oxygen consumption rate (OCR) following the manufacturer's protocol. The glycolysis of cardiomyocytes was measured using Seahorse XF Glycolysis Stress Test Kit (103020-100, Agilent Technologies) by measuring the extracellular acidification rate (ECAR) according to the manufacturer's instruction.

## Echocardiography

Transthoracic echocardiography was performed using the Vevo 1100 Ultrasound machine (VisualSonics, Toronto, ON, Canada). A month after surgery, mice were anesthetized and placed on the heated platform. Left ventricular ejection fraction (LVEF) and fractional shortening (FS) were determined. All measurements were performed according to the American Society for Echocardiography leading-edge technique standards. Three consecutive cardiac cycles were analyzed and the average was used for data analysis.

## Histology and immunofluorescence analysis

Mouse hearts were fixed with 4% paraformaldehyde solution, dehydrated, and embedded in paraffin. 4 μm thick sections were obtained for staining. Tissues were stained with hematoxylin and eosin (HE) (G1120, Solarbio) and the Masson kit (G1340, Solarbio) according to the manufacturer's instructions. Scar size was measured using the Image Pro Plus software.

For immunofluorescence staining, heart tissues were embedded in OCT compound and sectioned at a thickness of 5 μm. Then tissue sections were fixed with acetone at 4 °C, penetrated with 0.5% TritonX for 30 min at room temperature, blocked at room temperature with goat serum (AR0009, Boster) for 10 min, and incubated with primary antibody at 4 °C overnight. After washing with PBS, tissues were incubated with secondary antibodies Alexa Fluor 488 (ab150113, Abcam, 1:400) and Alexa Fluor 594 (ab150080, Abcam, 1:400) at room temperature for 60 min in the dark. Thereafter, tissues were stained with DAPI (C0065, Solarbio, 1:500) at room temperature for 15 min. Images were captured using a fluorescent microscope (Olympus, JAPAN) and analyzed with ImageJ software.

## Statistics

Data are expressed as mean ± SEM. Data distribution was evaluated by the Kolmogorov-Smirnov test. An F-test was used to evaluate the homogeneity of variance. Normally distributed data with only 1 variable were analyzed by parametric analysis: an unpaired (2-tailed) Student's t-test (with Welch correction when variance was unequal) for 2 groups and 1-way analysis of variance (ANOVA) with post-hoc Tukey or Dunnet test for more than two groups. Non-normally distributed data with only 1 variable were analyzed by nonparametric analysis: Mann-Whitney U test (2-tailed) for 2 groups and 1-way analysis of variance (ANOVA) with a Kruskal-Wallis test for more than two groups. Data with more than 1 variable were evaluated by 2-way ANOVA, with Bonferroni post-tests.

## Reporting summary

Further information on research design is available in the Nature Portfolio Reporting Summary linked to this article.

## Data availability

The sequencing raw data generated in this study have been deposited in Sequence Read Archive (SRA) at the NCBI Center with the accession number PRJNA974152. All other data generated in this study are provided in the Source data file. The Source data are provided with this paper.

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

## Acknowledgements

This work was supported by the National Natural Science Foundation of China (92168119 and 82373958 to B.C., 82100300 to W.M., 82304480 to X.W.), the Natural Science Foundation of Heilongjiang Province (YQ2020H015 to W.M.), the HMU Marshal Initiative Funding (HMUMIF-21018 to B.C.).

## Author contributions

B.C. and W.M. conceived the study concept. W.M., YN.T., L.S., H.C., H.S., H.J., X.L., W.H., XL.G., X.J., X.W., Y.L., Y.Y., XF.G., Q.OY. and JL.L. performed the experimental studies. W.M., YN.T. and L.S. carried out the data analysis. B.C., J.L., Y.T., F.Y., F.L., N.W. and W.M. wrote the manuscript. B.C., W.M. and X.W. provided the funding. All authors reviewed the manuscript. W.M., YN.T. and L.S. contributed equally to this work.

## Competing interests

The authors declare no competing interests.
