## [Peer Review File · Nature Communications]

N-Acetyltransferase 10 represses Uqcr11 and Uqcrb independently of ac4C modification to promote heart regenerationReviewer #1 (Remarks to the Author):

In this study, the authors investigate molecular mechanisms of heart regeneration in the neonatal heart by RNA- and ribosome profiling/Ribo-seq to identify and test key molecular players placing translational regulation as critical in this process. Finding translational efficiency of OXPHOS-associated genes to be downregulated; Uqcr11, Uqcrb, and Atp5j2 are confirmed by changes in protein but not mRNA levels. Testing candidates regulating these genes the authors isolate Nat10 as a translational regulator specifically. Nat10 is revealed to suppress translation of Uqcr11 and Uqcrb to alter mitochondrial function and facilitate regeneration. Interestingly, Nat10 does not act through ac4C modification of Uqcr11/Uqcrb mRNAs, confirmed with expression of an enzyme-dead Nat10 mutant. Associating levels of Nat10 with cardiac development and cardiomyocyte cell cycle regulation, the authors also show cardiomyocyte-specific loss of Nat10 inhibits cardiac regeneration in vivo, while gain of Nat10 enhances this. Moving to larger mammals, Nat10 expression decreases over neonatal development in pigs and is reduced in failing human hearts. Finally, use of Remodilin to inhibit Nat10, limits cardiac regeneration. The current study is compelling and extensive, but several technical concerns preclude publication at this time as detailed below.

- 1) Although a semi-quantitative technique, accurate western blotting quantification is key to this study where a central finding that changes in protein expression occur that do not correlate with mRNA. It is understandable that proteins less than 20 kDa can be difficult to resolve by western blotting, but it is not possible to perform densitometry on blots where bands have joined together (as in Figures 3F, 3G, 4F, 6A, 6G, 7F, and 9I for example). How was densitometry performed here? The authors need to use a higher percentage gels to better resolve these proteins and quantify.
- 2) Uqcr11 in Figure 3G is clearly not a 16 kDa band from the same gel as Uqcrb and Atp5j2 and would therefore require presentation independently with a separate loading control. The authors need to provide whole uncropped images of the blots used to generate this Figure. Nowhere else in the manuscript is Uqcr11 presented as such clearly defined bands for such a small protein, which raises significant concerns regarding rigor that need to be addressed.
- 3) Westerns in Figure 7F are also concerning. Tubulin is labeled at 37 kDa, is this GAPDH? Even so, why are bands running into each other at this larger MW? Again, whole uncropped images of blots that generated these figures need to be provided.
- 4) How is RNA localization/nuclear export assessed (Figure 4C)? some representative raw data and not just summary data should be presented here.
- 5) What is the sample size for polysome profiling experiments? Are these polysome profiling data significant and consistent across reps?
- 6) Minor but important: 'western' blotting is not named after an individual, and so should not be capitalized.

Reviewer #2 (Remarks to the Author):

Review for the manuscript NCOMMS-23-20604 entitled "N-Acetyltransferase 10 represses the expression of Uqcr11 and Uqcrb independently of ac4C modification to promote heart regeneration"

The authors present a study investigating the role of N-Acetyltransferase 10 (Nat10) in cardiac regeneration and its underlying mechanism. They report that Nat10 acts as a key regulator mediating the translation of OXPHOS-related genes Uqcr11 and Uqcrb independent of its ac4C enzyme activity. They demonstrate that Nat10 overexpression promotes cardiac regeneration and enhances cardiac function after injury while inhibition of Nat10 impairs heart regeneration and repair in neonatal mice. Based on these findings the authors propose Nat10 as a crucial

translational regulator during heart regeneration with potential clinical applications. However, the mechanism of translational regulation by Nat10 remains unknown. The crucial activity of NAT10 for translational regulation of Uqcr11 and Uqcrb is not addressed. Specifically, the specific binding of NAT10 to Uqcr11 and Uqcrb mRNAs needs validation using the NAT10 mutant that is defective in mRNA binding along with the identification of the cis-elements in Uqcr11 and Uqcrb mRNAs crucial for recognition by NAT10. To be considered for publication in Nature Communications, crucial results supporting their proposal must be obtained.

Comments:

1) The authors analyze the translation efficiency of Uqcr11 Uqcrb and Atp5j2 using polysome profiling but fail to provide A254 absorbance data to indicate the position of 40S 60S 80S and polysomes. Including this information is crucial to validate the observed translation changes in the target mRNAs.

2) In the discussion (line 438), the authors suggest that Nat10 affects translation and expression by binding to the mRNA of Uqcrb and Uqcr11 blocking their nuclear export and reducing the number of translatable mRNAs. However, experimental evidence supporting the specific binding of NAT10 to Uqcr11 and Uqcrb mRNAs must be provided. The NAT10 mutant which is defective in the binding to Uqcr11 and Uqcrb mRNAs must be isolated and used as a negative control for RIP assay together with Atp5j2 mRNA. Moreover, the identification of cis-elements in Uqcr11 and Uqcrb mRNAs crucial for recognition by NAT10 is necessary.

3) Another major concern is regarding the activity of Nat10 and its inhibition by Remodelin. The authors demonstrate that an ac4C modification-defective Nat10 mutant (Nat10-G614E) still regulates the translation of Uqcr11 and Uqcrb and they also show that Remodelin impairs cardiomyocyte proliferation and cardiac repair. However, since Remodelin is not a specific inhibitor of NAT10-catalyzed RNA acetylation (PMID: 34141066), it is necessary to investigate the activity of NAT10 inhibited by Remodelin. Deletion analysis of NAT10 function in translation regulation should be performed and the effect of Nat10-G614E on protein acetylation should be evaluated. Additionally, since acetylation at Lys-426 is required for the activation of rRNA transcription it is essential to investigate the activity of NAT10-K426R in the translational regulation of Uqcr11 and Uqcrb mRNAs.

Minor comments:

It is recommended to describe the functions of Uqcr11 Uqcrb and Atp5j2 in the introduction to give readers a better understanding of their relevance to the study. Furthermore, in Figure 4D please include the amino acid numbers in the mutated region to provide clearer information.

Reviewer #3 (Remarks to the Author):

In this manuscript, authors found that translation of the genes related to oxidative phosphorylation were down-regulated in heart regeneration. After integrated analysis of RNA, RPF and TE data, authors focused on the OXPHOS-related genes Uqcr11, Uqcrb and Atp5j2, which were down-regulated in cardiac regeneration. Further, authors found that these genes translation were controlled by Nat10. Additionally, authors found that Nat10 regulates the translation of Uqcr11 and Uqcrb to control cardiomyocyte proliferation in an ac4C-independent manner. By using cardiomyocyte-specific genetic Nat10 overexpression or Nat10 knockout mice model, authors showed that Nat10 promotes heart regeneration in neonatal mice. Therefore, authors claimed that Nat10 is a key translational regulator during heart regeneration and has promising prospects for clinical application. However, there is no direct evidence supporting that Nat10 controls protein translation as authors found that Nat10 blocks the nuclear export of Uqcr11 and Uqcrb mRNA instead of inhibiting translation directly. As known, protein translation occurs in the cytoplasm. In addition, to confirm that Nat10 inhibits the translation of Uqcr11 and Uqcrb, pulse-chase experiments are required to verify the level of newly synthesized Uqcr11 and Uqcrb proteins.

Major points,

1. When explore the key regulators of translation for Uqcr11, Uqcrb and Atp5j2, should authors

analyze the interaction of the mRNAs of Uqcr11, Uqcrb and Atp5j2, with the proteins that could be the potential translation regulators? Why did authors analyze the PPI network (Figure 3B)?

2. In Fig. 5A, authors showed Nat10 is upregulated in the AR group, how authors explain the upregulation of Nat10? This should be discussed.

3. In the manuscript, authors used a single siRNA targeting Nat10. To exclude the off-target effect, at least two pairs of Nat10 siRNA are required to verify the effect of Nat10 on the cardiac regeneration and translational regulation. The same with Uqcr11 siRNA and Uqcrb siRNA.

4. Remodelin is known as an inhibitor of Nat10, which inhibits the acetyltransferase activity without affecting Nat10 expression level. Fig. 9C shows that Remodelin decreased the Nat10 expression level, why? Authors should make sure they used correct reagent and also in a proper dosage by evaluating the acetylation of Nat10 substrates.

5. Fig. 9I, the Western blot of Uqcr11 didn't show clear band. A Western blot in good quality is required.

6. Fig 6C, D, E, the labels of histograms are not legible.

7. Authors claimed that Remodelin promoted the translation and protein expression of Uqcr11 and Uqcrb (Fig 9H and I), what is the mechanism involved? Remodelin is a specific inhibitor of NAT10, does Nat10 control protein translation via its lysine acetyltransferase activity?

8. To address that Nat10 inhibits the nuclear export of Uqcr11 and Uqcrb mRNA, in situ hybridization of Uqcr11 and Uqcrb mRNA in the Nat10-overexpressed cells is required.

9. In this manuscript, the upregulation of Nat10 is the key, it is necessary to explain how Nat10 is up-regulated in cardiac regeneration.

REVIEWER COMMENTS

Reviewer #1 (Remarks to the Author):

In this study, the authors investigate molecular mechanisms of heart regeneration in the neonatal heart by RNA- and ribosome profiling/Ribo-seq to identify and test key molecular players placing translational regulation as critical in this process. Finding translational efficiency of OXPHOS-associated genes to be downregulated; Uqcr11, Uqcrb, and Atp5j2 are confirmed by changes in protein but not mRNA levels. Testing candidates regulating these genes the authors isolate Nat10 as a translational regulator specifically. Nat10 is revealed to suppress translation of Uqcr11 and Uqcrb to alter mitochondrial function and facilitate regeneration. Interestingly, Nat10 does not act through ac4C modification of Uqcr11/Uqcrb mRNAs, confirmed with expression of an enzyme-dead Nat10 mutant. Associating levels of Nat10 with cardiac development and cardiomyocyte cell cycle regulation, the authors also show cardiomyocyte-specific loss of Nat10 inhibits cardiac regeneration in vivo, while gain of Nat10 enhances this. Moving to larger mammals, Nat10 expression decreases over neonatal development in pigs and is reduced in failing human hearts. Finally, use of Remodilin to inhibit Nat10, limits cardiac regeneration. The current study is compelling and extensive, but several technical concerns preclude publication at this time as detailed below.

Reply: We would like to express our sincere gratefulness for your positive comments. Your suggestions are constructive and helpful for improving the quality of our manuscript. We have carefully read the comments and made the corrections according to the suggestions. Additional experiments with changing experimental conditions and improving experimental techniques have been performed to improve the quality of the results. As suggested, all the full western blot images have been provided, and also uploaded as a related manuscript file in the revised manuscript. We hope that all the corrections and new experimental results are able to address the questions you have raised. Please find our point-by-point reply to your individual comments and suggestions below.

1) Although a semi-quantitative technique, accurate western blotting quantification is key to this study where a central finding that changes in protein expression occur that do not correlate with mRNA. It is understandable that proteins less than 20 kDa can be difficult to resolve by western blotting, but it is not possible to perform densitometry on blots where bands have joined together (as in Figures 3F, 3G, 4F, 6A, 6G, 7F, and 9I for example). How was densitometry performed here? The authors need to use a higher percentage gels to better resolve these proteins and quantify.

Reply: Thank you for your valuable comments. We agree with your opinion that accurate western blotting quantification is very important. Actually, in the original data, we performed the densitometry of adjacent bands by zooming in on the image and segmenting them based on the centerline between the lanes. Nevertheless, just as you mentioned, small molecular weight proteins less than 20 kDa are difficult to resolve, and the bands that have joined together may interfere with the densitometry. In order to avoid this interference, we repeated these relevant experiments by using high percentage gels according to your advice and increasing the distance between the lanes, and the bands that joined together have replaced by the new ones. The consistent experimental conclusions have been reached as before, and the full western blot images we obtained were provided as shown below.

Western blot images for Figure 3F, 3G, 4F, 6A, 6G, 7F, and 9I.

Figure 3F and Figure S4

The full western blot images for Figure 3F and Figure S4.

Figure 3G

The full western blot images for Figure 3G.

Figure 4F

The full western blot images for Figure 4F.

Figure 6G

The full western blot images for Figure 6G.

Figure 7F

The full western blot images for Figure 7F.

Figure 9I

The full western blot images for Figure 9I.

2) Uqcr11 in Figure 3G is clearly not a 16 kDa band from the same gel as Uqcrb and Atb5j2 and would therefore require presentation independently with a separate loading control. The authors need to provide whole uncropped images of the blots used to generate this Figure. Nowhere else in the manuscript is Uqcr11 presented as such clearly defined bands for such a small protein, which raises significant concerns regarding rigor that need to be addressed.

Reply: We appreciate your comments. Yes, the bands of Uqcr11 in Figure 3G were indeed not from the same gel as Uqcrb and Atp5j2. According to your suggestion, we provided the whole uncropped images of the blots used to generate this figure as below. The bands were occasionally obtained by one student. We also have repeated the experiments during the past three months and obtained the same conclusion. To maintain the consistency of the figures, we replaced the old blots with the new images (as shown below). Moreover, according to your comments, we have presented the western blot results with their respective loading controls.

Figure 3G

old

new

The full western blot images for the old and new Figure 3G.

3) Westerns in Figure 7F are also concerning. Tubulin is labeled at 37 kDa, is this GAPDH? Even so, why are bands running into each other at this larger MW? Again, whole uncropped images of blots that generated these figures need to be provided.

Reply: Thanks for your nice questions. (1) Yes, you are right. The band is Tubulin and the molecular weight should be 55 kDa, which was mistakenly labeled as 37 kDa. We are very sorry for this, and this error has been corrected. (2) As suggested, we provided the whole uncropped images of Figure 7F as shown below. The reasons for the bands running into each other in Figure 7F might be a large sample loading amount and a long time running during electrophoresis. To demonstrate the better images, we performed these experiments again, and the new images were obtained as shown below. We replaced the old image with a new one in Figure 7F.

Figure 7F

old

new

The full western blot images for the old and new Figure 7F.

4) How as RNA localization/nuclear export assessed (Figure 4C)? some representative raw data and not just summary data should be presented here.

Reply: We appreciate your question and comment. This comment is very constructive in improving the quality of our manuscript. (1) We are sorry not to introduce this clearly in the original manuscript. In our study, the RNA localization was assessed by detecting the expression of RNA in the cytoplasm and nucleus via nuclear and cytoplasmic separation assays (Figure A). We have included the detailed information about this method in the revised manuscript. To further support our findings, we employed FISH to assess the Uqcrb and Uqcr11 mRNAs localization in cardiomyocytes. The results showed that Nat10 overexpression increased the nuclear distribution of Uqcrb and Uqcr11 mRNAs (Figure B). These data suggest that Nat10 overexpression induces the retention of Uqcrb and Uqcr11 mRNAs in the nucleus. Additionally, we further performed the experiment to reveal how Nat10 affects the nuclear export of Uqcr11 and Uqcrb. The Nuclear RNA Export Factor 1 (Nxf1) is the main 'transport vehicle' for mRNAs. Thus, we detected if Nat10 affects the interaction of mRNAs with Nxf1. The result of Nxf1 RIP experiment showed that overexpression of Nat10 decreased the abundance of Uqcr11 and Uqcrb mRNAs precipitated by Nxf1 (Figure C). The data described above implies that Nat10 binds to Uqcr11 and Uqcrb mRNAs, inhibiting the interaction between their mRNAs and Nxf1, thereby affecting their nuclear export. (2)

According to your comments, we have changed the presentation of the graph, which features information about the distribution of the underlying data (Figure A). The raw data has been included in the table in the Source data file (also shown below).

The distribution of Uqcr11 and Uqcrb between the nucleus and cytoplasm and the interaction of Uqcr11 and Uqcrb mRNAs with the Nuclear RNA Export Factor 1 (Nxf1). **A** The total RNA was extracted from cytoplasm and nucleus and the mRNA of Uqcr11 and Uqcrb was detected using qRT-PCR (n=3). **B** The RNA localization of Uqcr11 and Uqcrb was assessed using FISH. **C** Nxf1 RIP analysis of Uqcr11 and Uqcrb (n=3). Two-tailed Student's t test with Welch's correction. Data are presented as mean \pm SEM.

The raw data of Figure 4C.

		18S		U6		Uqcrb		Uqcr11	
		CT value	Ratio	CT value	Ratio	CT value	Ratio	CT value	Ratio
Vector Nucleus	1	11.48221	0.060399	12.43891	0.849846	22.33659	0.574754	23.30103	0.441711
	2	11.47626	0.054159	12.57887	0.83141	22.12808	0.466869	23.41241	0.416379
	3	11.69709	0.044338	12.53168	0.844639	22.20072	0.449639	23.13059	0.467017
Nat10 OE Nucleus	1	10.6828	0.145961	11.49641	0.939278	21.75923	0.822363	22.66026	0.752182
	2	10.6402	0.149869	11.53071	0.934328	21.95076	0.800424	22.93049	0.687517
	3	10.66176	0.131784	11.35492	0.936758	21.9133	0.831577	22.84139	0.707633
Vector Cytoplasm	1	7.522757	0.939601	14.93967	0.150154	22.77123	0.425246	22.96312	0.558289
	2	7.349936	0.945841	14.88091	0.16859	21.93661	0.533131	22.92528	0.583621
	3	7.267201	0.955662	14.97439	0.155361	21.90911	0.550361	22.93998	0.532983
Nat10 OE Cytoplasm	1	8.134079	0.854039	15.44766	0.060722	23.97007	0.177637	24.26206	0.247818
	2	8.136219	0.850131	15.36129	0.065672	23.95459	0.199576	24.06811	0.312483
	3	7.941882	0.868216	15.24364	0.063242	24.21706	0.168423	24.11661	0.292367

5) What is the sample size for polysome profiling experiments? Are these polysome profiling data significant and consistent across reps?

Reply: Thank you for your nice question. Indeed, we have conducted three or four repeated experiments and these polysome profiling data was significant and consistent across replicates. As suggested, we have updated the graph to fully and clearly exhibit the data, and the *P*-values for statistical significance are shown below.

Figure 2I

Fractions	P value (AR vs. sham)			Fractions	P value (AR vs. sham)		
	Uqcr11	Uqcrb	Atp52		Uqcr11	Uqcrb	Atp52
1	0.02	0.007	0.50	6	0.02	0.001	0.008
2	0.007	0.04	0.88	7	0.06	0.03	0.16
3	0.02	0.06	0.33	8	0.009	0.06	0.21
4	0.02	0.02	0.53	9	0.004	0.03	0.04
5	0.01	0.03	>0.99	10	0.99	0.03	0.15

Figure 3E

Fractions	P value (Nat10 OE vs. Vector)			P value (Akt9D OE vs. Vector)		
	Uqcr11	Uqcrb	Atp52	Uqcr11	Uqcrb	Atp52
1	0.04	0.02	0.18	0.25	0.44	0.19
2	0.002	<0.001	0.43	0.22	0.87	0.15
3	0.06	0.01	0.14	0.08	0.58	0.30
4	0.07	0.09	0.77	0.08	0.23	>0.99
5	<0.001	0.02	0.45	0.56	0.69	0.38
6	0.003	0.005	0.34	0.12	0.97	0.22
7	0.02	0.36	0.07	0.45	0.65	0.18
8	0.009	0.05	0.86	0.20	0.01	0.37
9	0.14	0.05	0.23	0.69	0.75	0.11
10	0.14	0.04	0.23	0.28	0.20	0.10

Figure 4G

Fractions	P value (Nat10 G641E vs. Vector)	
	Uqcr11	Uqcrb
1	0.41	0.003
2	<0.001	0.008
3	0.10	0.54
4	0.13	0.29
5	0.37	0.06
6	0.30	0.31
7	0.06	0.18
8	0.03	0.05
9	0.18	0.05
10	0.10	0.14

Figure 5F

Fractions	P value (Myh6/Nat10 Flox+ vs. Nat10 Flox+)	
	Uqcr11	Uqcrb
1	0.02	0.001
2	0.001	<0.001
3	0.05	0.32
4	0.004	0.60
5	0.41	0.010
6	0.003	0.004
7	0.51	0.18
8	0.02	0.01
9	0.07	0.02
10	0.43	0.02

Figure 7E

Fractions	P value (Nat10 KI vs. WT)	
	Uqcr11	Uqcrb
1	0.001	0.003
2	0.01	0.001
3	0.01	0.005
4	<0.001	0.80
5	0.007	0.03
6	0.01	0.01
7	0.005	0.006
8	0.31	0.010
9	0.65	0.20
10	0.02	0.33

Figure 8C

Fractions	P value (P7 vs. P1)	
	Uqcr11	Uqcrb
1	0.04	0.02
2	0.05	0.03
3	0.001	0.05
4	0.09	0.02
5	0.70	0.12
6	0.005	0.09
7	0.02	0.14
8	0.02	0.05
9	0.25	0.10
10	<0.001	0.008

Figure 8G

Fractions	P value (Nat10 OE vs. Vector)	
	Uqcr11	Uqcrb
1	0.76	0.46
2	0.002	0.03
3	0.02	0.04
4	0.03	0.14
5	0.06	0.008
6	0.08	0.007
7	0.28	0.01
8	0.006	0.02
9	0.18	0.008
10	0.004	0.008

Figure 9H

Fractions	P value (Remodulin vs. Vehicle)	
	Uqcr11	Uqcrb
1	0.006	0.05
2	0.10	0.02
3	0.07	0.11
4	0.01	<0.001
5	<0.001	0.15
6	0.12	0.02
7	0.002	0.07
8	0.04	0.07
9	0.002	0.03
10	0.01	0.003

The Polysome profiling analyses and the relevant P values.

6) Minor but important: 'western' blotting is not named after an individual, and so should not be capitalized.

Reply: Thank you for your kind reminder. According to your advice, "Western blot" has been replaced by "western blot" throughout the manuscript, except when it appears at the beginning of a sentence. The corrections have been marked in the revised manuscript.

Reviewer #2 (Remarks to the Author):

Review for the manuscript NCOMMS-23-20604 entitled "N-Acetyltransferase 10 represses the expression of Uqcr11 and Uqcrb independently of ac4C modification to promote heart regeneration"

The authors present a study investigating the role of N-Acetyltransferase 10 (Nat10) in cardiac regeneration and its underlying mechanism. They report that Nat10 acts as a key regulator mediating the translation of OXPHOS-related genes Uqcr11 and Uqcrb independent of its ac4C enzyme activity. They demonstrate that Nat10 overexpression promotes cardiac regeneration and enhances cardiac function after injury while inhibition of Nat10 impairs heart regeneration and repair in neonatal mice. Based on these findings the authors propose Nat10 as a crucial translational regulator during heart regeneration with potential clinical applications. However, the mechanism of translational regulation by Nat10 remains unknown. The crucial activity of NAT10 for translational regulation of Uqcr11 and Uqcrb is not addressed. Specifically, the specific binding of NAT10 to Uqcr11 and Uqcrb mRNAs needs validation using the NAT10 mutant that is defective in mRNA binding along with the identification of the cis-elements in Uqcr11 and Uqcrb mRNAs crucial for recognition by NAT10. To be considered for publication in Nature Communications, crucial results supporting their proposal must be obtained.

Reply: We would like to express our sincere thanks for your constructive comments. Your suggestions are helpful in strengthening our findings and improving the quality of this manuscript. To address the concerns you pointed out, we have performed the following experiments to reveal the mechanism underlying the translational regulation of Uqcr11 and Uqcrb by Nat10. Firstly, in order to reveal if Nat10 ac4C function is involved in this process, we constructed a mutation of Nat10 in the 641st amino acid residue (Nat10 G641E) which loses its ac4C modification ability on mRNAs. The results demonstrated that Nat10 G641E still has the ability to suppress the translation of Uqcr11 and Uqcrb in cardiomyocytes, which is the same as the wild-type Nat10, indicating that Nat10 regulation of Uqcr11 and Uqcrb is independent of ac4C modification of their mRNAs. Besides, our study further uncovered that Nat10 was able to bind to Uqcr11 and Uqcrb mRNAs directly and alter their subcellular localization by antagonizing their interaction with Nxf1, the main 'transport vehicle' for mRNAs. In addition, we also constructed the Nat10 mutations defective in the binding sites for Uqcr11 and Uqcrb mRNAs, and found that the Nat10 mutations lose the ability to inhibit the nuclear export and protein expression of Uqcr11 and Uqcrb mRNAs in cardiomyocytes. Furthermore, we constructed the biotin-labeled mutant Uqcr11 and

Uqcrb mRNAs without the binding sequences for Nat10, and then RNA pull-down assays identified the specific sequences of Uqcr11 and Uqcrb mRNAs binding to wild-type Nat10. The results reveal the key binding sites and the mRNA sequences for Nat10 interacting with Uqcr11 and Uqcrb mRNAs. All these findings support that Nat10 plays an inhibitory role in Uqcr11 and Uqcrb expression by binding to their mRNAs specifically and affecting their cytoplasmic distribution. The detailed responses to the comments are shown below.

Comments:

1) The authors analyze the translation efficiency of Uqcr11 Uqcrb and Atp5j2 using polysome profiling but fail to provide A254 absorbance data to indicate the position of 40S 60S 80S and polysomes. Including this information is crucial to validate the observed translation changes in the target mRNAs.

Reply: Thank you for your kind reminder. The absorbance data can indicate the positions of 40S, 60S, 80S, and polysomes, and is capable to display the overall translation profiling. According to your advice, we have included all the A254 absorbance data in the supplementary file of the revised manuscript (also as shown below). Although the absorbance data can indicate the overall translation levels, the translation changes of the specific mRNAs need to be confirmed by qRT-PCR using their specific primers. Thus, in our study, we detected the translation of target mRNAs by measuring their distribution among different sucrose gradient fractions. The results were included in the Figure 2I, 3E, 4G 6F, 7E, 8C, 8G, and 9H in the revised manuscript.

The A254 absorbance data of Polysome profiling analyses.

2) In the discussion (line 438), the authors suggest that Nat10 affects translation and expression by binding to the mRNA of Uqcrb and Uqcr11 blocking their nuclear export and reducing the number of translatable mRNAs. However, experimental evidence supporting the specific binding of NAT10 to Uqcr11 and Uqcrb mRNAs must be provided. The NAT10 mutant which is defective in the binding to Uqcr11 and Uqcrb mRNAs must be isolated and used as a negative control for RIP assay together with Atp5j2 mRNA. Moreover, the identification of cis-elements in Uqcr11 and Uqcrb mRNAs crucial for recognition by NAT10 is necessary.

Reply: Thank you for your valuable comments. As suggested, we have conducted the following experiments to further reveal the specific binding of Nat10 to Uqcr11 and Uqcrb mRNAs. Firstly, we predicted the binding sites between Nat10 and Uqcr11 or Uqcrb using the catRAPID algorithm. The prediction results showed that 224-277 amino acid residues of Nat10 have the highest potential to bind with Uqcr11, and 724-775 amino acid residues have the highest score for binding to Uqcrb (Figure A and B). Then, we constructed the Nat10 plasmids without 224-277 (Nat10- Δ 224-277) or 724-775 (Nat10- Δ 724-775) residues (Figure C), and then detected the interaction of Uqcr11 and Uqcrb mRNAs with the mutant Nat10 using RIP experiment. The RIP experiment revealed that the wild-type Nat10 could precipitate the mRNAs of Uqcr11 and Uqcrb,

while Nat10- Δ 224-277 was only able to precipitate Uqcrb mRNA with significantly decreased abundance of Uqcr11, and Nat10- Δ 724-775 still could precipitate Uqcr11 mRNA but at an obviously reduced level of Uqcrb. However, Atp5j2 could not be precipitated by either the wild-type or mutant Nat10 (Figure C). Furthermore, we also found that Nat10- Δ 224-277, which loses the ability to interact with Uqcr11, had no inhibitory effect on Uqcr11 mRNA's nuclear export and protein expression, and Nat10- Δ 724-775, which cannot interact with Uqcrb, lost the inhibitory effect on Uqcrb mRNA's nuclear export and protein expression (Figure D-F). These results reveal the key amino acid residues of Nat10 interacting with Uqcr11 and Uqcrb mRNAs in cardiomyocytes.

On the other hand, bioinformatic analysis indicates that the most likely binding sites of Uqcr11 mRNA to Nat10 protein are the 136-184 and 161-212 sequences of Uqcr11 mRNA, and the potential binding sites of Uqcrb to Nat10 are the 426-477 and 151-202 sequences of Uqcrb mRNA (Figure A and B). Therefore, we constructed the biotin-labeled wild-type and mutant Uqcr11 and Uqcrb mRNAs (Figure G). RNA pull-down experiment displayed that the wild-type Uqcr11 and Uqcrb mRNAs could pull down Nat10, while the mutants of Uqcr11 in the 136-187 and 161-212 sequences, or the mutants of Uqcrb in the 151-202 and 426-477 sequences, lost or weakened their interaction with Nat10 (Figure G). Furthermore, we blasted the binding sequences 136-187 and 161-212 of Uqcr11 mRNA, and 151-202 and 426-477 of Uqcrb mRNA, and found no similar sequences as the cis-elements in Uqcr11 and Uqcrb mRNAs for recognition by Nat10 (Figure H). These results indicate that Nat10 specifically binds to Uqcr11 and Uqcrb mRNAs, respectively.

At last, we further explored the mechanism by which Nat10 affects the nuclear export of Uqcr11 and Uqcrb mRNA. The Nuclear RNA Export Factor 1 (Nxf1) is the main 'transport vehicle' for mRNAs. We investigated if Nat10 affects the interaction of these mRNAs with Nxf1. The results showed that overexpression of Nat10 decreased the abundance of Uqcr11 and Uqcrb mRNAs pulled down by Nxf1 (Figure I). These results imply that Nat10 specifically binds to the Uqcr11 and Uqcrb mRNAs, inhibiting their interaction with Nxf1, and thereby affecting their nuclear export.

The identification of the specific interaction of Nat10 with Uqcr11 and Uqcrb. A and B The prediction of the binding sites between Nat10 and Uqcr11 or Uqcrb using the catRAPID algorithm. **C** The interaction of mutant Nat10 with Uqcr11 and uqcrb mRNA (n=3). **D** The distribution of Uqcr11 and Uqcrb between the nucleus and cytoplasm (n=3). **E and F** Western blot analysis of Uqcrb and Uqcr11. **G** The interaction of mutant Uqcr11 and Uqcrb mRNAs with Nat10 protein. **H** The sequences of the binding sites of Uqcr11 and Uqcrb. **I** The interaction of Uqcr11 and Uqcrb mRNAs with the Nuclear RNA Export Factor 1 (Nxf1) (n=3). Two-tailed Student's t test. Data are presented as mean ± SEM.

3) Another major concern is regarding the activity of Nat10 and its inhibition by Remodelin. The authors demonstrate that an ac4C modification-defective Nat10 mutant (Nat10-G614E) still regulates the translation of Uqcr11 and Uqcrb and they also show that Remodelin impairs cardiomyocyte proliferation and cardiac repair. However, since Remodelin is not a specific inhibitor of NAT10-catalyzed RNA acetylation (PMID:

34141066), it is necessary to investigate the activity of NAT10 inhibited by Remodelin. Deletion analysis of NAT10 function in translation regulation should be performed and the effect of Nat10-G614E on protein acetylation should be evaluated. Additionally, since acetylation at Lys-426 is required for the activation of rRNA transcription it is essential to investigate the activity of NAT10-K426R in the translational regulation of Uqcr11 and Uqcrb mRNAs.

Reply: Thanks a lot for your good advice. (1) Our study revealed that Nat10 regulates the translation of Uqcr11 and Uqcrb mRNAs independently of ac4C modification of their mRNAs. To further confirm this, we constructed a mutant of the Nat10 acetyltransferase activity site (Nat10-G641E), and detected the effect of Nat10-G641E on protein acetylation and mRNA ac4C modification. Western blot results showed that the overexpression of wild-type Nat10 increases the level of acetyl- α -tubulin K40, a known Nat10 lysine acetylase substrate, but Nat10-G641E failed to do this function (Figure A). The ac4C RIP analysis showed that overexpression of wild-type Nat10 increases the ac4C modification of Tfec mRNA, which can be subjected to ac4C modification by Nat10 (PMID: 35138696), but Nat10-G641E failed to do this function (Figure B). The data indicate that Nat10-G641E loses its acetyltransferase activity. Nevertheless, we found that Nat10-G641E could still affect the translation of Uqcr11 and Uqcrb in cardiomyocytes just as wild-type Nat10 (Figure C). It further confirms that Nat10 regulates the translation of Uqcr11 and Uqcrb independently of ac4C modification and protein acetylation.

(2) To further support our findings, we performed the following studies to better clarify the role of ac4C modification in the regulation of Uqcr11 and Uqcrb mRNAs by Remodelin. We overexpressed wild-type Nat10 and the Nat10 with acetyltransferase active site mutation (Nat10-G641E) in cardiomyocytes, and detected the effect of Remodelin on the expression of Uqcr11 and Uqcrb proteins, and the level of acetyl- α -tubulin K40. The results showed that Remodelin has similar effects on the expression of Uqcr11 and Uqcrb protein in both wild-type Nat10 and Nat10-G641E groups, but caused different effects on the acetyl- α -tubulin K40 level (Figure D). It indicates Remodelin exerts its effects independently of its inhibitory effect on acetylation activity. Recent studies have reported that Remodelin was able to inhibit the expression level of Nat10 (PMID: 35522942, PMID: 36522719, PMID: 36939377, PMID: 36765042, PMID: 35743017, PMID: 35978804). Consistent with these studies, our research also found that Remodelin reduced the level of Nat10 protein in cardiomyocytes. To further clarify whether Remodelin regulates cardiomyocyte proliferation and heart repair through the inhibition of the Nat10 expression, affecting the interaction of Nat10 with Uqcr11 and Uqcrb mRNA, we also investigated the effects of Remodelin on the expression of Uqcr11 and Uqcrb protein, and the level of acetyl- α -tubulin K40 in wild-

type Nat10 and the mutant Nat10 (Nat10- Δ 224-277 and Nat10- Δ 724-775) transfected cardiomyocytes, which loses their ability to binding with Uqcr11 or Uqcrb mRNAs. The results showed that Remodelin produces different effects on the expression of Uqcr11 and Uqcrb proteins between wild-type Nat10 and the mutant Nat10 groups, but does not induce the difference in the level of acetyl- α -tubulin K40 between the two groups (Figure E). It means that Remodelin regulates Uqcr11 and Uqcrb expression through Nat10 interacting directly with their mRNAs, thus regulating cardiomyocyte proliferation and cardiac repair.

(3) According to your comment, we further investigated the role of acetylation at Lys-426 of Nat10 in the translational regulation of Uqcr11 and Uqcrb. We constructed the plasmid of Nat10-K426R and performed polysome profiling assay in cardiomyocytes. The results showed that wild-type Nat10 increased the absorbance of 40S compared to the vector, indicating that overexpression of wild-type Nat10 promotes rRNA transcription; whereas, the Nat10-K426R has no difference from the control group in the absorbance of 40S, indicating that the mutant Nat10 at K426 lost its function in the activation of rRNA transcription (Figure F). Then, RT-PCR results showed that Nat10-K426R decreased the translation of Uqcr11 and Uqcrb compared to the control group. However, the inhibitory effect was not further enhanced by the mutants compared to the wild-type Nat10 group (Figure F). Additionally, we demonstrated that the binding sites of Nat10 interacting with Uqcr11 and Uqcrb are 224-277 and 724-775 amino acid residues (as shown in the second comment). This suggests that Nat10-K426R still can affect the translation of Uqcr11 and Uqcrb, as the mutation at the 426st amino acid residue does not affect the interaction of Nat10 with their mRNAs. These data indicate that although acetylation at Lys-426 is required for the activation of rRNA transcription, the regulatory effect of Nat10 on the translation of Uqcr11 and Uqcrb is not achieved by affecting rRNA transcription, rather than by binding to their mRNAs and affecting their abundance in the cytoplasm, as we demonstrated.

(4) At last, in the original results, we have provided the data that overexpression of Nat10 inhibited the translation of Uqcr11 and Uqcrb. As suggested, we performed the experiment to study the effect of Nat10 knockdown on the translational regulation of Uqcr11 and Uqcrb. The results showed that interfering with the expression of Nat10 promotes the translation of Uqcr11 and Uqcrb (Figure G).

The effects of Remodelin on acetyltransferase activity of Nat10 and the effect of Nat10 siRNA, Nat10-K426R on the translational regulation of Uqcr11 and Uqcrb mRNAs. **A** Western blot detects the expression of acetyl- α -tubulin K40. **B** Ac4C RIP analysis of Uqcr11 and Uqcrb. **C** Polysome profiling analysis of Uqcr11 and Uqcrb in cardiomyocytes transfected with Nat10-G641E. **D** and **E** The effect of Remodelin on the expression of ac- α -tubulin, Uqcr11 and Uqcrb. **F** and **G** Polysome profiling analysis of Uqcr11 and Uqcrb in cardiomyocytes transfected with Nat10-K426R and Nat10 siRNA.

Minor comments:

It is recommended to describe the functions of Uqcr11, Uqcrb and Atp5j2 in the introduction to give readers a better understanding of their relevance to the study. Furthermore, in Figure 4D please include the amino acid numbers in the mutated region to provide clearer information.

Reply: We apologize for the missing description of Uqcr11, Uqcrb and Atp5j2. The detail description has been added into the "Introduction" of the revised manuscript. Accordingly, we also included the amino acid numbers in the mutated region in Figure 4D to provide clearer information.

Reviewer #3 (Remarks to the Author):

In this manuscript, authors found that translation of the genes related to oxidative phosphorylation were down-regulated in heart regeneration. After integrated analysis of RNA, RPF and TE data, authors focused on the OXPHOS-related genes Uqcr11, Uqcrb and Atp5j2, which were down-regulated in cardiac regeneration. Further, authors found that these genes translation were controlled by Nat10. Additionally, authors found that Nat10 regulates the translation of Uqcr11 and Uqcrb to control cardiomyocyte proliferation in an ac4C-independent manner. By using cardiomyocyte-specific genetic Nat10 overexpression or Nat10 knockout mice model, authors showed that Nat10 promotes heart regeneration in neonatal mice. Therefore, authors claimed that Nat10 is a key translational regulator during heart regeneration and has promising prospects for clinical application. However, there is no direct evidence supporting that Nat10 controls protein translation as authors found that Nat10 blocks the nuclear export of Uqcr11 and Uqcrb mRNA instead of inhibiting translation directly. As known, protein translation occurs in the cytoplasm. In addition, to confirm that Nat10 inhibits the translation of Uqcr11 and Uqcrb, pulse-chase experiments are required to verify the level of newly synthesized Uqcr11 and Uqcrb proteins.

Reply: Thank you for your constructive comments. (1) Our study found that Nat10 affected the translation and expression of Uqcr11 and Uqcrb, but did not observe the ac4C modification on the mRNA of Uqcr11 and Uqcrb (Figure A). The mutation of Nat10 acetyltransferase activity site still altered the translation of Uqcr11 and Uqcrb (Figure B). It indicates that Nat10 regulates the expression of Uqcr11 and Uqcrb independently of ac4C modification. RIP assays further uncovered that Nat10 was able to bind to Uqcr11 and Uqcrb mRNAs (Figure C). This sparked our interest. Subsequently, we found that although the binding of Nat10 to Uqcr11 and Uqcrb did not affect their mRNA levels, it could alter their subcellular localization (Figure D and E). Then, we conducted the following study to further confirm if Nat10 affects the nuclear export of Uqcr11 and Uqcrb mRNAs by inhibiting the interaction of mRNAs with the Nuclear RNA Export Factor 1 (Nxf1), the main 'transport vehicle' for mRNA (Figure F). The results showed that overexpression of Nat10 decreased the abundance of Uqcr11 and Uqcrb mRNAs pulled down by Nxf1, leading to less translatable mRNAs in the cytoplasm, thus affecting their translation. This finding further supports the translation regulation of Uqcrb and Uqcr11 by Nat10 from a new perspective. Moreover, we further conducted the experiments to prove the specific binding of Nat10 to Uqcr11 and Uqcrb mRNAs. We predicted the binding sites between Nat10 and Uqcr11 or Uqcrb using the catRAPID algorithm. The prediction results show that 224-277 amino acid residues of Nat10 has a greater potential to bind with Uqcr11, and 724-775 amino acid

residues has a highest score for binding to Uqcrb (Figure G and H). Subsequent analysis revealed that the most probable binding sites of Uqcr11 mRNA to Nat10 protein are the 136-184 and 161-212 sequences of Uqcr11 mRNA, and the highest potential binding sites of Uqcrb to Nat10 are the 426-477 and 151-202 sequences of Uqcrb mRNA (Figure G and H). Thus, we constructed the Nat10 overexpression plasmids without 224-277 (Nat10- Δ 224-277) or 724-775 (Nat10- Δ 724-775) residues (Figure I), and detected the interaction of Uqcr11 and Uqcrb mRNAs with the mutant Nat10 using RIP experiment. The RIP experiment revealed that the wild-type Nat10 could precipitated the mRNAs of Uqcr11 and Uqcrb, while the Nat10- Δ 224-277 was only able to precipitate Uqcrb mRNA with significantly decreased abundance of Uqcr11, and Nat10- Δ 724-775 still could precipitate Uqcr11 mRNA but at an obviously reduced level of Uqcrb; However, Atp5j2 could not be precipitated by either the wild-type or mutant Nat10 (Figure I). Furthermore, we found that Nat10- Δ 224-277, which was unable to interact with Uqcr11, no longer had an inhibitory effect on Uqcr11 mRNA's nuclear export and protein expression, and Nat10- Δ 724-775, which did not interact with Uqcrb, lost the inhibitory effect on Uqcrb mRNA's nuclear export and protein expression (Figure J-L). Moreover, we also constructed the biotin-labeled wild-type and mutant Uqcr11 and Uqcrb mRNAs (Figure M). RNA pull-down experiment displayed that the wild-type Uqcr11 and Uqcrb mRNAs could pull down Nat10, while the mutants of Uqcr11 in the 136-187 and 161-212 sequences, and of Uqcrb in the 151-202 and 426-477 sequences, either lost or weakened their interaction with Nat10 (Figure M). These results indicate that Nat10 specifically binds to Uqcr11 and Uqcrb mRNAs, respectively. Collectively, these data suggest Nat10 specifically binds to the Uqcr11 and Uqcrb mRNAs, inhibiting the interaction between them and Nxf1, and thereby affecting their nuclear export and protein expression. We included these results into the revised manuscript.

Nat10 regulates the expression of Uqcr11 and Uqcrb by binding to their mRNAs specifically. **A** Ac4C RIP analysis of Uqcr11 and Uqcrb. **B** Polysome profiling analysis of Uqcrb and Uqcr11 in mutant Nat10 overexpressed cardiomyocytes. **C** Nat10 RIP analysis of Uqcr11 and Uqcrb. **D** The distribution of Uqcr11 and Uqcrb between the nucleus and cytoplasm. **E** The RNA localization of Uqcr11 and Uqcrb assessed using FISH. **F** The interaction of Uqcr11 and Uqcrb mRNAs with the Nuclear RNA Export Factor 1 (Nxf1). **G** and **H** The prediction of the binding sites between Nat10 and Uqcr11 or Uqcrb using the catRAPID algorithm. **I** The interaction of mutant Nat10 with Uqcr11 and uqcrb mRNA (n=3). **J** The distribution of Uqcr11 and Uqcrb between the nucleus and cytoplasm (n=3). **K** and **L** Western blot analysis of Uqcrb and Uqcr11. **M** The

interaction of mutant Uqcr11 and Uqcrb mRNAs with Nat10 protein. Two-tailed Student's t test. Data are presented as mean \pm SEM.

(2) We have thought to do the pulse-chase experiments to confirm if Nat10 inhibits the levels of newly synthesized Uqcr11 and Uqcrb. This experiment requires the use of radioactive Methionine, whereas our laboratory has no permission to use radioactive elements. The radioactive materials are not accessible under the current condition. But we want to address the concern as possible as we can. We performed the following experiments to test that Nat10 inhibited the levels of newly synthesized Uqcr11 and Uqcrb. Firstly, we applied MG132 to inhibit the protein degradation pathway in Nat10 overexpressing cardiomyocytes. The results showed that, after inhibiting the protein degradation pathway, overexpression of Nat10 could still inhibit the protein levels of both Uqcr11 and Uqcrb (Figure A). It indicates that Nat10 does not affect the degradation of Uqcr11 and Uqcrb proteins. Then, we found that Nat10 did not affect the mRNA levels of Uqcr11 and Uqcrb but changed their protein levels by qRT-PCR and western blot (Figure B-D). It is well known that gene expression is related to RNA transcription, mRNA decay, translation, and protein degradation. The mRNA levels and protein degradation of Uqcr11 and Uqcrb are not regulated by Nat10. So, the regulation of Uqcr11 and Uqcrb by Nat10 should be at the translational level. Meanwhile, the polysome profiling analysis also showed that Nat10 regulated the translation of Uqcr11 and Uqcrb (Figure E). The data described above support that Nat10 controls the translation of Uqcr11 and Uqcrb in cardiomyocytes.

Nat10 regulates the expression of Uqcr11 and Uqcrb. **A** The effect of Nat10 on the degradation of Uqcr11 and Uqcrb. **B** The effects of Nat10 on the mRNA expression of Uqcrb, Uqcr11 and Atp5j2 measured by qRT-PCR. **C** and **D** Western blot analysis of Nat10, Uqcrb, Uqcr11 and Atp5j2. **E** Polysome profiling analysis of Uqcrb, Uqcr11 and Atp5j2 in Nat10-overexpressed cardiomyocytes

Major points,

1. When explore the key regulators of translation for Uqcr11, Uqcrb and Atp5j2, should authors analyze the interaction of the mRNAs of Uqcr11, Uqcrb and Atp5j2, with the proteins that could be the potential translation regulators? Why did authors analyze the PPI network (Figure 3B)?

Reply: Thank you for your question. Generally, protein-protein interaction refers to a physical interaction, that is, binding or chemical reaction between proteins through spatial conformation or chemical bonds, which is the main research object of protein-protein interaction. But in addition to physical interactions, protein-protein interactions

also include genetic interactions, which refer to the interaction between proteins or coding genes under the influence of other proteins or genes in special environments, often manifested as phenotypic changes. We used the STRING database (<https://string-db.org/>) and Cytoscape (v. 3.7.1) to retrieve and analyze the potential interactions between the genes that regulate translation and our target genes. The relationships here include physical contact and targeted regulation. Therefore, we predicted the interaction between them from multiple perspectives and identified the potential and key regulatory gene as Nat10. Of course, we understand your concerns, so in order to avoid causing confusion for you and other readers, we have modified the description in the original manuscript.

2. In Fig. 5A, authors showed Nat10 is upregulated in the AR group, how authors explain the upregulation of Nat10? This should be discussed.

Reply: This is a really good suggestion. As the transcriptional level of Nat10 is upregulated in the AR group compared to the sham group, therefore we searched for the vital transcription factors regulating Nat10 expression. Hes1 was predicted to be potential to bind to the promoter region of Nat10 by both PROMO (https://alggen.lsi.upc.es/cgi-bin/promo_v3/promo/promoinit.cgi?dirDB=TF_8.3) and JASPAR (<http://jaspar.genereg.net/>) coupled with the UCSC Genome Browser (<http://genome.ucsc.edu>). Four binding sites were predicted in the Nat10 promoter region (Figure A). To confirm the regulatory effect of Hes1 on Nat10 in cardiomyocytes, we constructed luciferase reporter vectors containing the full-length promoter of Nat10. The dual-luciferase reporter assays revealed that overexpression of Hes1 led to a promotion of Nat10 promoter activity (Figure B). Sequential deletions of the Nat10 promoter, according to the presumptive binding sites, uncovered that the deletion of the Nat10 promoter containing fragment of 1500-1509 resulted in the disappearance of Hes1 overexpression activation of Nat10 transcriptional activity (Figure B). Subsequently, we detected the effect of Hes1 on Nat10 expression. The results showed that overexpression of Hes1 promoted, while Hes1 siRNA reduced the expression of Nat10 (Figure C). We also observed the expression of Hes1 in P1, P7, and AR hearts. The results showed that Hes1 is decreased in P7 heart compared to P1 heart, while increased in AR hearts compared to sham (Figure D and E). These results imply that the reason for the upregulation of Nat10 is attributed to the upregulation of its transcription factor Hes1.

The expression of Nat10 is regulated by Hes1. **A** The prediction of Hes1 binding to the promoter region of Nat10 by both PROMO (https://algggen.lsi.upc.es/cgi-bin/promo_v3/promo/promoinit.cgi?dirDB=TF_8.3) and JASPAR (<http://jaspar.genereg.net/>) coupled with UCSC Genome Browser (<http://genome.ucsc.edu>). **B** The luciferase assay indicates the interaction of Hes1 with the promoter of Nat10. **C** The effects of Hes1 knockdown and overexpression on Nat10 protein expression. **D** The expression of Nat10 in P1 and P7 heart tissues. **E** The expression of Nat10 in hearts of sham and AR mice.

3. In the manuscript, authors used a single siRNA targeting Nat10. To exclude the off-target effect, at least two pairs of Nat10 siRNA are required to verify the effect of Nat10 on the cardiac regeneration and translational regulation. The same with Uqcr11 siRNA and Uqcrb siRNA.

Reply: Thank you for your valuable comments. Indeed, we have constructed three pairs of siRNAs for each gene. For Nat10 and Uqcrb, two pairs of siRNAs (Nat10 siRNA-1: GCAUUCGGGUAUCCAAUATT, UAUUGGAAUACCCGAAUGCTT; Nat10 siRNA-2: CCCUCAGUCCUGUGGUGUUTT, AACACCACAGGACUGAGGGTT; Uqcrb siRNA-1: GCAAGUGGCUGGAUGGUUTT, AAACCAUCCAGCCACUUGCTT; Uqcrb siRNA-2:

CCUAAGGAUCAGUGGACAATT, UGUCCACUGAUCCUUAGGTT) have been shown to successfully knock down the mRNA expression (Figure A and B). Thus, we included the data of the other pairs of siRNAs into the revised manuscript. For Uqcr11 and Atp5j2, only one pair of siRNA works (Uqcr11 siRNA-1: AGUUUAAGAAGGACGAUUATT, UAAUCGUCCUUCUAAACUTT; Atp5j2 siRNA-1: GUACAUCAACGUUCGGAAATT, UUCCGAACGUUGAUGUACTT) (Figure C and D). So, we have constructed and screened the other effective siRNAs for Uqcr11 and Atp5j2 (Uqcr11 siRNA-2: ACAUCAACGGCAAGUUUAATT, UAAACUUGCCGUUGAUGUTT; Atp5j2 siRNA-2: CUGCAUUUCUACAAGGAATT, UCCUUGUAAGAAAUGCAGTT) (Figure E and F). We performed the experiments to test the effects of the new siRNAs against Uqcr11 and Atp5j2 on cardiomyocyte proliferation. The results showed that two pairs of Nat10, Uqcrb, Uqcr11 and Atp5j2 have the same impact on cardiac regeneration: Knockdown of Nat10 attenuated the proliferation of cardiomyocytes, and knockdown of Uqcr11 and Uqcrb promoted cardiomyocyte proliferation. But Atp5j2 siRNA had no effect on cardiomyocyte proliferation (Figure G and H). As suggested, we included this data into the revised manuscript.

The effects of Nat10, Uqcrb, Uqcr11 and Atp5j2 siRNA on cardiomyocyte

proliferation. A-F the transfection efficiency of Nat10, Uqcrb, Uqcr11 and Atp5j2 siRNAs. **G-H** The effects of Nat10, Uqcrb, Uqcr11 and Atp5j2 siRNA on cardiomyocyte proliferation analyzed by the detection of EdU, pH3 and Ki67.

4. Remodelin is known as an inhibitor of Nat10, which inhibits the acetyltransferase activity without affecting Nat10 expression level. Fig. 9C shows that Remodelin decreased the Nat10 expression level, why? Authors should make sure they used correct reagent and also in a proper dosage by evaluating the acetylation of Nat10 substrates.

Reply: Thanks for your nice comment. Actually, Remodelin was first identified as capable of inhibiting the activity of Nat10's lysine acetyltransferase (PMID: 24786082), which was cited by many following studies (PMID: 29703891, PMID: 31616951, PMID: 35967285, PMID: 34513300). However, the specific inhibitory action of Remodelin on the acetyltransferase activity of Nat10 is challenged by recent studies. It was found no direct evidence for interaction of Remodelin with the Nat10 acetyltransferase active site (PMID: 34141066). Interestingly, increasing evidence from recent studies reported that Remodelin was able to inhibit the expression level of Nat10 (PMID: 35522942, PMID: 36522719, PMID: 36939377, PMID: 36765042, PMID: 35743017, PMID: 35978804), suggesting Remodelin at least as an inhibitor of Nat10 expression. Consistent with these studies, our research also found that Remodelin reduced the level of Nat10 protein in cardiomyocytes, which also may explain why Remodelin regulates cardiomyocyte proliferation and heart repair in an ac4C-independent manner. The dosage of Remodelin in our study was similar to or smaller than that used in other studies to inhibit Nat10 activity (PMID: 35967285, PMID: 35743017, PMID: 36522719, PMID: 37237981). The reagent we used was consistent with other labs (PMID: 37328448, PMID: 36609449, PMID: 36939377, PMID: 31722219). In addition, we also evaluated the acetylation of Nat10 substrates. It was reported that the Tfec mRNA, as a substrate of Nat10, can be subjected to ac4C modification by Nat10 (PMID: 35138696). Therefore, we tested the effect of Remodelin on Tfec mRNA ac4C modification using ac4C RIP experiment. The results showed that Remodelin treatment caused the reduction of ac4C modification of the Tfec mRNA (Figure A), which is possibly attributed to the reduction of Nat10 expression by Remodelin. In addition, α -tubulin is known as a Nat10 lysine acetylase substrate, so we examined the acetyl- α -tubulin K40 levels. The results showed that the acetyl- α -tubulin K40 level was inhibited after treatment with Remodelin (Figure B). To better clarify the role of ac4C modification in the regulation of Uqcr11 and Uqcrb mRNAs by Remodelin, we transfected cardiomyocytes with wild-type Nat10 and the acetylation-defective Nat10 mutant (Nat10-G641E), and observed the effect of Remodelin on the expression of Uqcr11 and Uqcrb protein, and the level of acetyl- α -tubulin K40. The results showed

that Remodelin had the similar effects on the expression of Uqcr11 and Uqcrb proteins in both wild-type Nat10 and Nat10-G641E groups, but caused the different effects on the acetyl- α -tubulin K40 level between wild-type Nat10 and Nat10-G641E groups (Figure C). It indicates Remodelin exerts its effects independent on its acetylation activity. Meanwhile, we also investigated the effects of Remodelin on expression of Uqcr11 and Uqcrb protein, and the level of acetyl- α -tubulin K40 in wild-type Nat10 and the mutants Nat10 that loses their ability to binding with Uqcr11 or Uqcrb mRNA. The results showed that Remodelin produced the different effects on the expression of Uqcr11 and Uqcrb protein between wild-type Nat10 and the mutant Nat10 groups, but there was no difference in the level of acetyl- α -tubulin K40 between wild-type Nat10 and the mutant Nat10 groups (Figure D). It means that Remodelin regulates Uqcr11 and Uqcrb expression through Nat10 interacting with their mRNAs directly. The above experimental results indicate that the application of Remodelin in our study is proper.

The mechanism of Remodelin regulates the expression of Uqcr11 and Uqcrb. A Ac4C RIP analysis of Uqcr11 (n=3) and Uqcrb (n=6). **B** Western blot detecting the expression of acetyl- α -tubulin K40. **C** and **D** The effect of Remodelin on the expression of ac- α -tubulin, Uqcr11 and Uqcrb.

5. Fig. 9I, the Western blot of Uqcr11 didn't show clear band. A Western blot in good quality is required.

Reply: Thank you for your nice comment. Your suggestion is very helpful to improve the quality of this manuscript. As suggested, we have replaced the unclear band with the better one (as shown below). We also provided the full western blot images of 9I as a related manuscript file in the revised manuscript.

Figure 9I

6. Fig 6C, D, E, the labels of histograms are not legible.

Reply: Thanks for your kind reminder. As suggested, we have relabeled the histograms to make it more legible (as shown below).

Figure 6C

Figure 6D

Figure 6E

7. Authors claimed that Remodelin promoted the translation and protein expression of Uqcr11 and Uqcrb (Fig 9H and I), what is the mechanism involved? Remodelin is a specific inhibitor of NAT10, does Nat10 control protein translation via its lysine acetyltransferase activity?

Reply: Thank you for your nice comments. Our study revealed that Remodelin promoted the translation and protein expression of Uqcr11 and Uqcrb in

cardiomyocytes by inhibiting the expression of Nat10. Consistent with our studies, many studies showed that Remodelin inhibited the expression of Nat10 protein in different cells (PMID: 36522719, PMID: 37237981, PMID: 35522942, PMID: 36939377, PMID: 36765042, PMID: 35978804). In addition, to detect whether Nat10 controls protein translation via its lysine acetyltransferase activity, we performed the following experiments. Because the 641st amino acid residue is vital for Nat10's acetyltransferase activity, we constructed the Nat10 G641E mutant plasmid and studied its effects on acetylation activity. The results showed that the mutant (Nat10 G641E) lost its acetylation activity against lysine (Figure A). We further observed the effects of the Nat10 G641E mutant on Uqcr11 and Uqcrb translation. The results showed that the Nat10 G641E mutant still could affect the translation of Uqcr11 and Uqcrb, indicating that Nat10 controls Uqcr11 and Uqcrb translation independently of its lysine acetyltransferase activity in this study (Figure B). Additionally, we further verified that Remodelin regulated the Uqcr11 and Uqcrb expression independent on the acetylation activity of Nat10. Our studies showed that Remodelin regulation of Uqcr11 and Uqcrb expression was weakened in cardiomyocytes transfected with the wild-type Nat10 compared with the vector group, but no changes in cardiomyocytes transfected with the mutant Nat10 that loses their ability to binding with Uqcr11 or Uqcrb mRNA (Figure C). Conversely, Remodelin regulation of Uqcr11 and Uqcrb expression was not affected between cardiomyocytes transfected with wild-type Nat10 and Nat10-G641E (Figure D). These findings suggest that Remodelin promotes the translation and protein expression of Uqcr11 and Uqcrb is through inhibiting Nat10 expression and its interaction with Uqcr11 and Uqcrb mRNAs, rather than affecting the acetyltransferase activity of Nat10.

The mechanism of Remodelin regulates the expression of Uqcr11 and Uqcrb. A Western blot detects the expression of acetyl- α -tubulin K40. **B** Polysome profiling

analysis of Uqcr11 and Uqcrb in cardiomyocytes transfected with Nat10-G641E. **C** and **D** The effect of Remodelin on the expression of ac- α -tubulin, Uqcr11 and Uqcrb.

8. To address that Nat10 inhibits the nuclear export of Uqcr11 and Uqcrb mRNA, in situ hybridization of Uqcr11 and Uqcrb mRNA in the Nat10-overexpressed cells is required.

Reply: We appreciate your valuable suggestion. According to your advice, we applied a Fluorescence in situ Hybridization (FISH) experiment to confirm the effects of Nat10 overexpression on the distribution of Uqcr11 and Uqcrb mRNA in cardiomyocytes. The results showed that Nat10 overexpression inhibited the expression levels of Uqcr11 and Uqcrb mRNA in the cytoplasm, but increased their level in the nucleus compared to control group (as shown below). This finding supports that Nat10 overexpression is capable of reducing the nuclear export of Uqcr11 and Uqcrb mRNA in cardiomyocytes.

The RNA localization of Uqcr11 and Uqcrb assessed using FISH.

9. In this manuscript, the upregulation of Nat10 is the key, it is necessary to explain how Nat10 is up-regulated in cardiac regeneration.

Reply: This is a really good question. According to your comments, we conducted the following experiments to clarify it. Because Nat10 was upregulated transcriptionally in this study, we thus focused on the potential transcription factors regulating Nat10 expression. By using PROMO database (https://algggen.lsi.upc.es/cgi-bin/promo_v3/promo/promoinit.cgi?dirDB=TF_8.3) and JASPAR (<http://jaspar.genereg.net/>) coupled with UCSC Genome Browser (<http://genome.ucsc.edu>), Hes1 was predicted as the potential transcription factor of Nat10. Hes1 was predicted to have four binding sites in the Nat10 promoter region (Figure A). To confirm transcriptional regulation of Nat10

by Hes1 in cardiomyocytes, we constructed luciferase reporter vectors containing the full-length promoter of Nat10. The dual-luciferase reporter assay revealed that the overexpression of Hes1 led to an increase of Nat10 promoter activity (Figure B). Sequential deletion of the Nat10 promoter region based on the presumptive binding sites, uncovered that the fragment of 1500-1509 in Nat10 promoter region is required for Hes1 transcriptional regulation of Nat10 (Figure B). Then, we further investigated the effect of gain and loss of function of Hes1 on Nat10 expression in cardiomyocytes. The results showed that knockdown of Hes1 by its siRNA reduced, while Hes1 overexpression by its plasmid promoted the expression of Nat10 (Figure C). Moreover, we also observed the change of Hes1 protein in P1, P7, and AR hearts. The results showed that the expression of Hes1 was decreased in P7 heart compared to P1 heart, while increased in AR hearts compared to sham (Figure D and E). These results imply that Nat10 is positively regulated by its transcription factor Hes1, and the upregulation of Nat10 is at least partially attributed to the upregulation of Hes1 in hearts.

The expression of Nat10 is regulated by Hes1. **A** The prediction of Hes1 binding to the promoter region of Nat10 by both PROMO (https://algggen.lsi.upc.es/cgi-bin/promo_v3/promo/promoinit.cgi?dirDB=TF_8.3) and JASPAR (<http://jaspar.genereg.net/>) coupled with UCSC Genome Browser (<http://genome.ucsc.edu>). **B** The luciferase assay indicates the interaction of Hes1 with

the promoter of Nat10. **C** The effects of Hes1 knockdown and overexpression on the expression of Nat10 protein. **D** The expression of Nat10 in P1 and P7 heart tissues. **E** The expression of Nat10 in hearts of sham and AR mice.

Reviewer #1 (Remarks to the Author):

The authors have made significant effort to address my concerns. Western blots have been improved significantly; RNA localization explained better, please italicize labeling of *Uqcr11* and *Uqcrb* in this figure to communicate better that it is RNA and not protein that is being labeled; Polysome quantification improved. I am satisfied with the work.

Reviewer #2 (Remarks to the Author):

Review for the manuscript NCOMMS-23-20604A entitled "N-Acetyltransferase 10 represses the expression of *Uqcr11* and *Uqcrb* independently of ac4C modification to promote heart regeneration"

The authors have addressed most of my previous concerns. I support the publication of manuscript in Nature Communications.

Reviewer #3 (Remarks to the Author):

I'm content with the authors' response. It is acceptable for publication.

REVIEWERS' COMMENTS

Reviewer #1 (Remarks to the Author):

The authors have made significant effort to address my concerns. Western blots have been improved significantly; RNA localization explained better, please italicize labeling of *Uqcr11* and *Uqcrb* in this figure to communicate better that it is RNA and not protein that is being labeled; Polysome quantification improved. I am satisfied with the work.

Reply: Thank you very much for your positive evaluation of our revised manuscript. Your suggestions are constructive and helpful for improving the quality of our manuscript. As requested, we have italicized *Uqcr11* and *Uqcrb* in the figure displaying their RNA localization, as shown below.

The distribution of *Uqcr11* and *Uqcrb* between the nucleus and cytoplasm. The RNA localization of *Uqcr11* and *Uqcrb* assessed using FISH.

Reviewer #2 (Remarks to the Author):

Review for the manuscript NCOMMS-23-20604A entitled "N-Acetyltransferase 10 represses the expression of *Uqcr11* and *Uqcrb* independently of ac4C modification to promote heart regeneration"

The authors have addressed most of my previous concerns. I support the publication of manuscript in Nature Communications.

Reply: Thank you for acknowledging our efforts to carefully address the concerns

raised. Thank you for your constructive comments once again.

Reviewer #3 (Remarks to the Author):

I'm content with the authors' response. It is acceptable for publication.

Reply: Thank you for the positive evaluation and recommendation of our work for publication in this journal.